# Cardiovascular disease risk factors induce mesenchymal features and senescence in mouse cardiac endothelial cells

**Karthik Amudhala Hemanthakumar[1,2], Shentong Fang[1,3], Andrey Anisimov[1,3], Mikko I Mäyränpää[4], Eero Mervaala[5], Riikka Kivelä[1,2]***

[1]Wihuri Research Institute, Helsinki, Finland; [2]Stem cells and Metabolism Research Program, Research Programs Unit, Faculty of Medicine, University of Helsinki, Helsinki, Finland; [3]Translational Cancer Medicine Research Program, Research Programs Unit, Faculty of Medicine, University of Helsinki, Helsinki, Finland; [4]Pathology, Helsinki University and Helsinki University Hospital, Helsinki, Finland; [5]Department of Pharmacology, Faculty of Medicine, University of Helsinki, Helsinki, Finland

**Abstract** Aging, obesity, hypertension, and physical inactivity are major risk factors for endothelial dysfunction and cardiovascular disease (CVD). We applied fluorescence-activated cell sorting (FACS), RNA sequencing, and bioinformatic methods to investigate the common effects of CVD risk factors in mouse cardiac endothelial cells (ECs). Aging, obesity, and pressure overload all upregulated pathways related to TGF-β signaling and mesenchymal gene expression, inflammation, vascular permeability, oxidative stress, collagen synthesis, and cellular senescence, whereas exercise training attenuated most of the same pathways. We identified collagen chaperone *Serpinh1* (also called as *Hsp47*) to be significantly increased by aging and obesity and repressed by exercise training. Mechanistic studies demonstrated that increased SERPINH1 in human ECs induced mesenchymal properties, while its silencing inhibited collagen deposition. Our data demonstrate that CVD risk factors significantly remodel the transcriptomic landscape of cardiac ECs inducing inflammatory, senescence, and mesenchymal features. SERPINH1 was identified as a potential therapeutic target in ECs.

*For correspondence:
riikka.kivela@helsinki.fi

Competing interests: The authors declare that no competing interests exist.

## Introduction

According to WHO, cardiovascular diseases (CVDs) account for 10% of the global disease burden and constitute the number 1 cause of death in the western world. CVDs are mainly caused by behavioral (physical inactivity, unhealthy diet) and metabolic (obesity, hypertension, diabetes, high cholesterol) risk factors (*Mendis et al., 2011*). Aging, however, is by far the biggest contributor to CVD, and aging population is becoming an enormous challenge worldwide.

The heart contains a dense vascular network, and endothelial cells (ECs) are indeed the most abundant cell population in the adult mouse heart (*Pinto et al., 2016*). In addition to their transport function, ECs are defined to control vasomotor tone, maintain vascular homeostasis, regulate angiogenesis, and establish bidirectional communication with other cell types and organs via paracrine signaling mechanisms (*Aird, 2007*; *Aird, 2012*; *Kivelä et al., 2019*; *Talman and Kivelä, 2018*; *Hemanthakumar and Kivelä, 2020*). ECs are found to be highly adaptive to physiological stimuli during normal growth and development (*White et al., 1998*; *Bloor, 2005*), and the diversity of ECs in different tissues has now been acknowledged. ECs are also maladaptive to a spectrum of pathological events involving, for example, inflammation or oxidative stress (*Cines et al., 1998*; *Gimbrone and García-Cardeña, 2016*), and the development of heart diseases is strongly linked to

**eLife digest** Cardiovascular diseases are the number one cause of death in the western world. Endothelial cells that line the blood vessels of the heart play a central role in the development of these diseases. In addition to helping transport blood, these cells support the normal running of the heart, and help it to grow and regenerate. Over time as the body ages and experiences stress, endothelial cells start to deteriorate. This can cause the cells to undergo senescence and stop dividing, and lay down scar-like tissue via a process called fibrosis. As a result, the blood vessels start to stiffen and become less susceptible to repair.

Ageing, obesity, high blood pressure, and inactivity all increase the risk of developing cardiovascular diseases, whereas regular exercise has a protective effect. But it was unclear how these different factors affect endothelial cells. To investigate this, Hemanthakumar et al. compared the gene activity of different sets of mice: old vs young, obese vs lean, heart problems vs healthy, and fit vs sedentary.

All these risk factors – age, weight, inactivity and heart defects – caused the mice's endothelial cells to activate mechanisms that lead to stress, senescence and fibrosis. Whereas exercise training had the opposite effect, and turned off the same genes and pathways. All of the at-risk groups also had high levels of a gene called SerpinH1, which helps produce tissue fiber and collagen. Experiments increasing the levels of SerpinH1 in human endothelial cells grown in the laboratory recreated the effects seen in mice, and switched on markers of stress, senescence and fibrosis.

According to the World Health Organization, cardiovascular disease now accounts for 10% of the disease burden worldwide. Revealing the affects it has on gene activity could help identify new targets for drug development, such as SerpinH1. Understanding the molecular effects of exercise on blood vessels could also aid in the design of treatments that mimic exercise. This could help people who are unable to follow training programs to reduce their risk of cardiovascular disease.

endothelial dysfunction and impaired vascular remodeling. However, the molecular cues, which cause maladaptation and dysfunction of ECs in the heart in response to pathological signals, remain elusive.

Physical inactivity increases the incidence of several chronic diseases, whereas regular exercise training has positive effects on most of our tissues (*Hawley et al., 2014*). Because microcirculation is present in every organ in the body, ECs have a unique ability to influence the homeostasis and function of different tissues, and they are potentially a major cell type mediating the positive effects of exercise throughout the body. Although the cardiac benefits of exercise are clear and there have been major advances in unraveling the molecular mechanisms, the understanding of how the molecular effects are linked to health benefits is still lacking (*Hawley et al., 2014*). Especially, the effects of exercise on ECs have not been characterized.

We hypothesized that the major CVD risk factors aging, obesity, and pressure overload will induce adverse remodeling of cardiac EC transcriptome (*Gimbrone and García-Cardeña, 2016*; *Ungvari et al., 2018*; *Brandes, 2014*), whereas exercise training would provide beneficial effects (*White et al., 1998*; *Bloor, 2005*). Both physiological and pathological stimuli significantly modified the cardiac EC transcriptome. Intriguingly, our results demonstrated that CVD risk factors promoted activation of transforming growth factor-β (TGF-β) signaling, inflammatory response, cellular senescence, and induced mesenchymal gene expression in cardiac EC, whereas exercise training promoted opposite protective effects.

## Results

### Exercise training and CVD risk factors modulate cardiac EC number, vascular density, and transcriptome

To mimic the effect of the most common CVD risk factors (aging, obesity, pressure overload/hypertension, and physical inactivity), we used adult C57BL/6J wild-type mice in the following experimental groups: aged (18 months) vs. young (2 months) mice, high-fat diet (HFD) induced obesity (14 weeks HFD) vs. lean mice, transverse aortic constriction (TAC) vs. sham-operated mice, and exercise

training (progressive treadmill running for 6 weeks) vs. sedentary mice (*Figure 1—figure supplement 1A,B*). Exercise trained mice showed improved ejection fraction compared to the sedentary mice, whereas aging, HFD, and TAC resulted in impaired heart function (*Figure 1—figure supplement 1C–F* and *Figure 1—source data 2*). HFD also induced marked weight gain, increased fat mass, and impaired glucose tolerance (*Figure 1—figure supplement 1G–I*). Left ventricular (LV) mass was increased in aged, HFD-treated, and TAC mice (*Figure 1—source data 2*). Exercise training also slightly increased LV mass, which reflects mild physiological hypertrophy often observed in endurance-trained athletes (*Arbab-Zadeh et al., 2014*; *Figure 1—source data 2*).

Exercise training significantly increased, whereas aging, HFD, and TAC decreased the percentage, count, and mean fluorescence intensity of the cardiac ECs (CD31$^+$CD140a$^-$CD45$^-$Ter119$^-$DAPI$^-$) compared to the controls, when analyzed by fluorescence-activated cell sorting (FACS; *Figure 1A,B*, *Figure 1—figure supplement 2A–D*). This was also demonstrated by immunohistochemistry for CD31-positive coronary vessels (*Figure 1C,D*). The cardiac ECs were gated and sorted by FACS (*Figure 1—figure supplement 3A*), and the isolated ECs were first analyzed by quantitative PCR analysis, which indicated significant enrichment of EC markers Cdh5 and Tie1 in the sorted fraction compared to whole heart or other cardiac mononuclear cells (*Figure 1—figure supplement 3B*). In addition, isolation resulted in 87.4 ± 1.9% cell viability and RNA purification strategy yielded intact and stable RNA with average RNA integrity number (RIN) of 8.7 (*Figure 1—figure supplement 3C, D*). RNA sequencing of isolated ECs was used to profile the expression pattern of cardiac EC transcripts in different experimental groups. Two-dimensional PCA of the EC transcriptomes exhibited significant proportion of variance in the gene expression pattern, which can be attributed to the treatment-induced changes in cardiac EC transcriptome (*Figure 2—figure supplement 1A–E*). Notably, unsupervised hierarchical clustering of EC data sets for all experimental interventions (sedentary, exercise trained, young, aged, sham, TAC) revealed consistent clustering and high degree of similarity in the gene expression pattern (*Figure 2—figure supplement 1F–J*). The analysis for differentially expressed genes (DEGs) showed a large number of up- and downregulated genes especially in aged, obese, and TAC-operated mice followed by a smaller number of affected genes in exercise trained mice. The number of significantly up- and downregulated genes with the false discovery rate (FDR) 0.05 for each treatment are shown in the MA plots and the top 50 DEGs for each treatment are presented by heat maps (*Figure 2A–E,F–J*).

## CVD risk factors induce senescence and TGF-β signaling together with mesenchymal gene expression in cardiac ECs

To understand the biological functions of the DEGs, we used PANTHER classification analysis (*Figure 3A*). The analysis revealed that genes related to EC development, adherence junction organization, IGFR signaling, adrenomedullin receptor signaling, and mitochondria were upregulated by exercise training. Furthermore, exercise training downregulated pathways related to cellular aging, vascular membrane permeability, negative regulation of angiogenesis, TGF-β1 production, collagen activated tyrosine kinase signaling, and ossification. In contrast, pathways related to TGF-β, IFNα, TNFα, oxidative stress, EC differentiation, vascular permeability, cell aging, collagen synthesis, SMAD signaling, and mesenchymal cell development were highly enriched in cardiac EC from both aged and obese mice. Downregulated pathways in these mice included tissue and lipid homeostasis, ECM assembly, tube morphogenesis, cell adhesion, cell number maintenance, EC proliferation, vasculature development, artery development, and NOTCH signaling. Pressure overload activated pathways such as cellular response to TGF-βR2 activation of fibrotic pathways, inactivation of cell survival pathways Erk1/2 and MAPK, and ossification process, whereas cellular homeostasis and vasculature development were repressed.

Comparison of the GO biological terms, which were significantly affected by exercise training and the CVD risk factors, demonstrated clear opposite effects on the EC transcriptome. Aging and HFD promoted oxidative stress response, activation of inflammatory and fibrosis pathways (*Figure 3—figure supplement 1A–E*) and cellular aging, and inhibited pathways regulating cell number maintenance, proliferation, and lipid homeostasis. Exercise training, in turn, promoted EC homeostasis and vascular growth, and prevented vascular aging, inflammation, and pathological activation. In the cardiac ECs of HFD and TAC-treated mice, a significant upregulation of senescence-associated secretory phenotype (SASP) genes (*Figure 3—figure supplement 2C–E*) were observed. To validate the link between obesity and senescence in the heart, we performed SA-β-galactosidase staining in

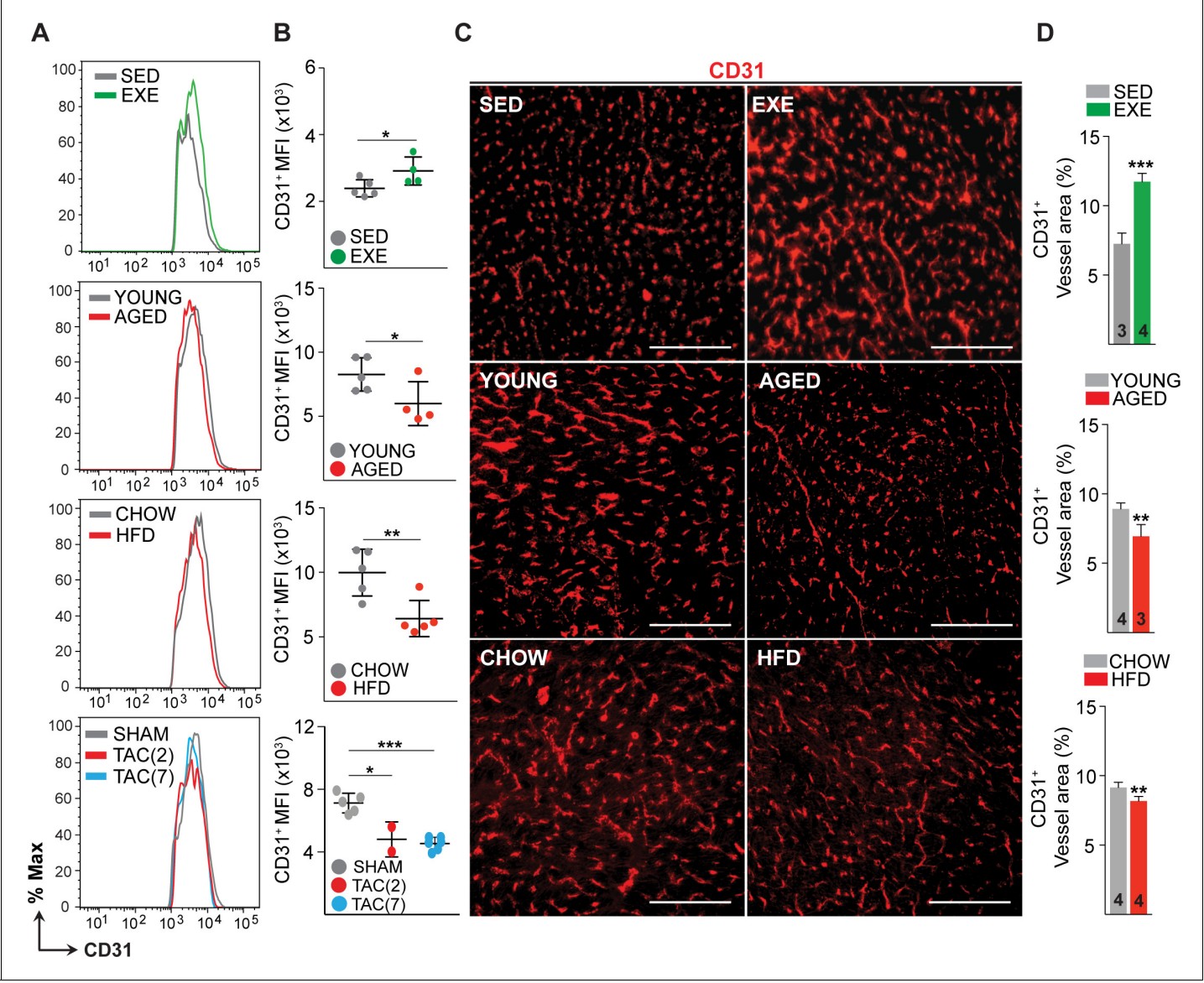

**Figure 1.** Effects of exercise training, aging, obesity, and pressure overload on cardiac endothelial cell (EC) number and vascular density.
(A and B) Fluorescence-activated cell sorting (FACS) analysis and quantification of mean fluorescence intensity (MFI) of the cardiac ECs (CD31+CD140a-CD45-Ter119-DAPI-) in various mouse models. (C and D) Representative immunofluorescence images and quantification of CD31+ blood vessel area (%) in the heart. Scale bar 100 µm. Data is presented as mean ± SEM. Student's t-test was used, *p<0.05, **p<0.01, ***p<0.001. In panel (B), each color-coded circle indicates an individual biological sample. In panel (D), the number of mice in each experimental group is indicated in the respective graph, N = 3–5 male mice/group.

The online version of this article includes the following source data and figure supplement(s) for figure 1:

**Source data 1.** Source data for *Figure 1B and D*.

**Source data 2.** Echocardiography measurements of cardiac function and ventricular dimensions in the indicated experimental group.

**Figure supplement 1.** Schematic of the experimental set-up to elucidate the impact of cardiovascular disease (CVD) risk factors on cardiac endothelial cell (EC) transcriptome and the validation of the experimental CVD risk factor models.

**Figure supplement 1—source data 1.** Source data for *Figure 1—figure supplement 1C, D, E, F, G, H and I*.

**Figure supplement 2.** Fluorescence-activated cell sorting (FACS) analysis of cardiac endothelial cell (EC).

**Figure supplement 2—source data 1.** Source data for *Figure 1—figure supplement 2A, B, C and D*.

**Figure supplement 3.** Quality metrics of the fluorescence-activated cell sorting (FACS) sorted cardiac endothelial cell (EC).

**Figure supplement 3—source data 1.** Source data for *Figure 1—figure supplement 3B*.

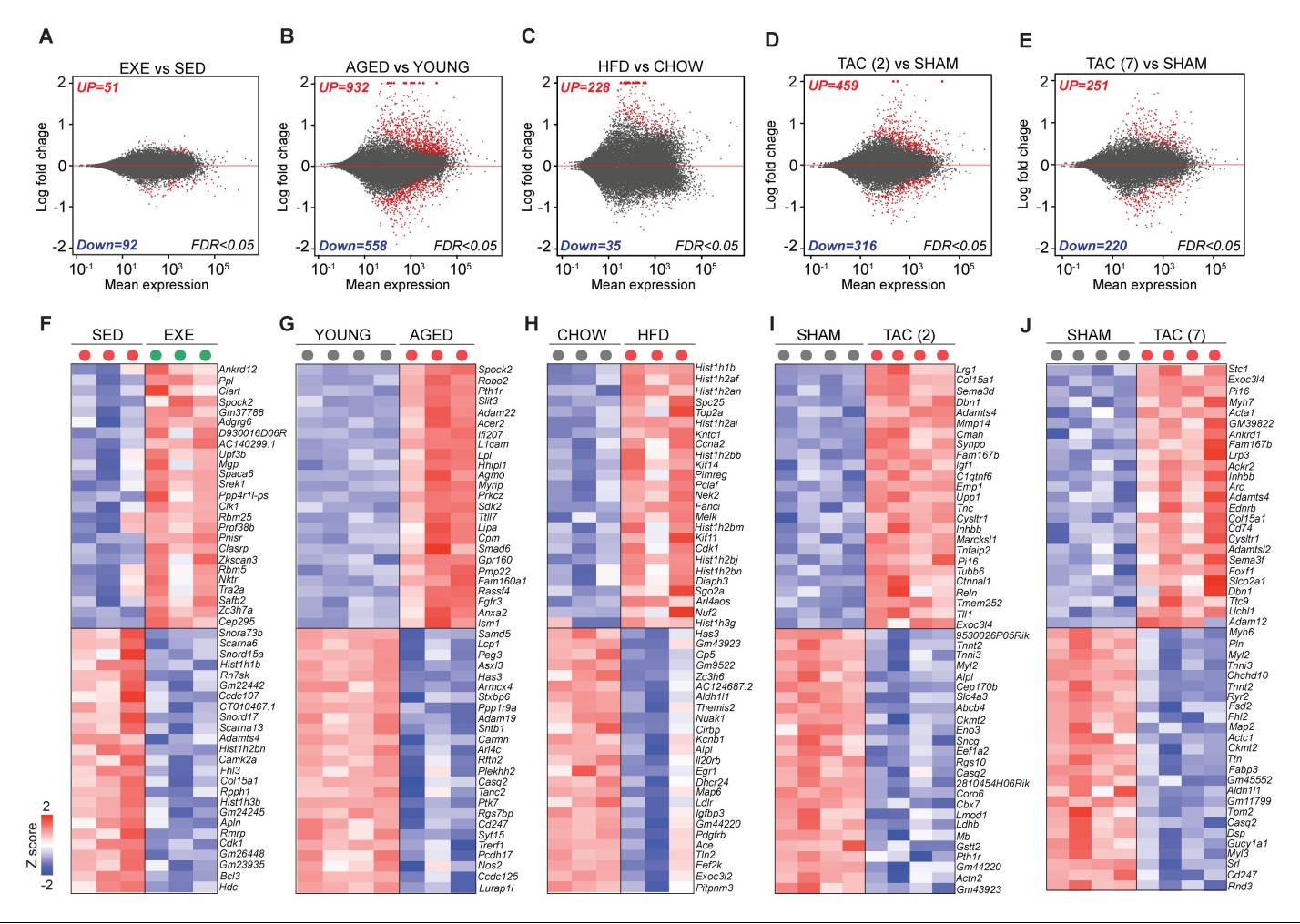

**Figure 2.** Transcriptomic changes in cardiac endothelial cells (ECs) in exercise trained, aged, obese, and transverse aortic constriction (TAC)-treated mice. (A–E) MA-plots (log ratio over mean) showing the number of differentially expressed genes (DEGs) in cardiac ECs for each experiment. Number of significantly up- and downregulated genes with the false discovery rate (FDR; Benjamini–Hochberg adjusted p-value) threshold of 0.05 are indicated in the plots. (F–J) Top 50 DEGs in cardiac ECs of the indicated experimental groups. In the heatmap, each color-coded circle (red, green, and black) indicates an individual biological sample within each experimental group. N = 3–4 male mice/group.

The online version of this article includes the following source data and figure supplement(s) for figure 2:

**Source data 1.** Source data for *Figure 2F, G, H, I and J*.

**Figure supplement 1.** Principal component analysis (PCA) plot and unsupervised hierarchical clustering of cardiac endothelial cell transcriptome from exercise trained, aged, obese, and transverse aortic constriction (TAC)-treated mice.

**Figure supplement 2.** Dispersion mean plot and p-value distribution plot of the indicated RNA sequencing experiments.

HFD- and chow-fed mouse heart sections. In HFD-fed mice, we observed several clusters of SA-β-galactosidase positive cells in the heart, which were not observed in the chow-fed animals (*Figure 3—figure supplement 2A*). Quantification showed significant increase in these cells (*Figure 3—figure supplement 2B*). Further studies are needed to identify these cells and their role in CVD development.

Because the analyses indicated upregulation of genes and pathways associated with mesenchymal development and endothelial-to-mesenchymal transition (EndMT) by all of the CVD risk factors, we reviewed our DEG sets for the expression of selected endothelial and mesenchymal markers based on the previously published data sets (*Figure 3—source data 1*). We found significant upregulation of many mesenchymal markers and downregulation of EC genes in aged and obese mice (*Figure 3C,D*). After 2 weeks of TAC, we also observed upregulation of several mesenchymal markers, whereas after 7 weeks of TAC, there was both up- and downregulation of the EC and

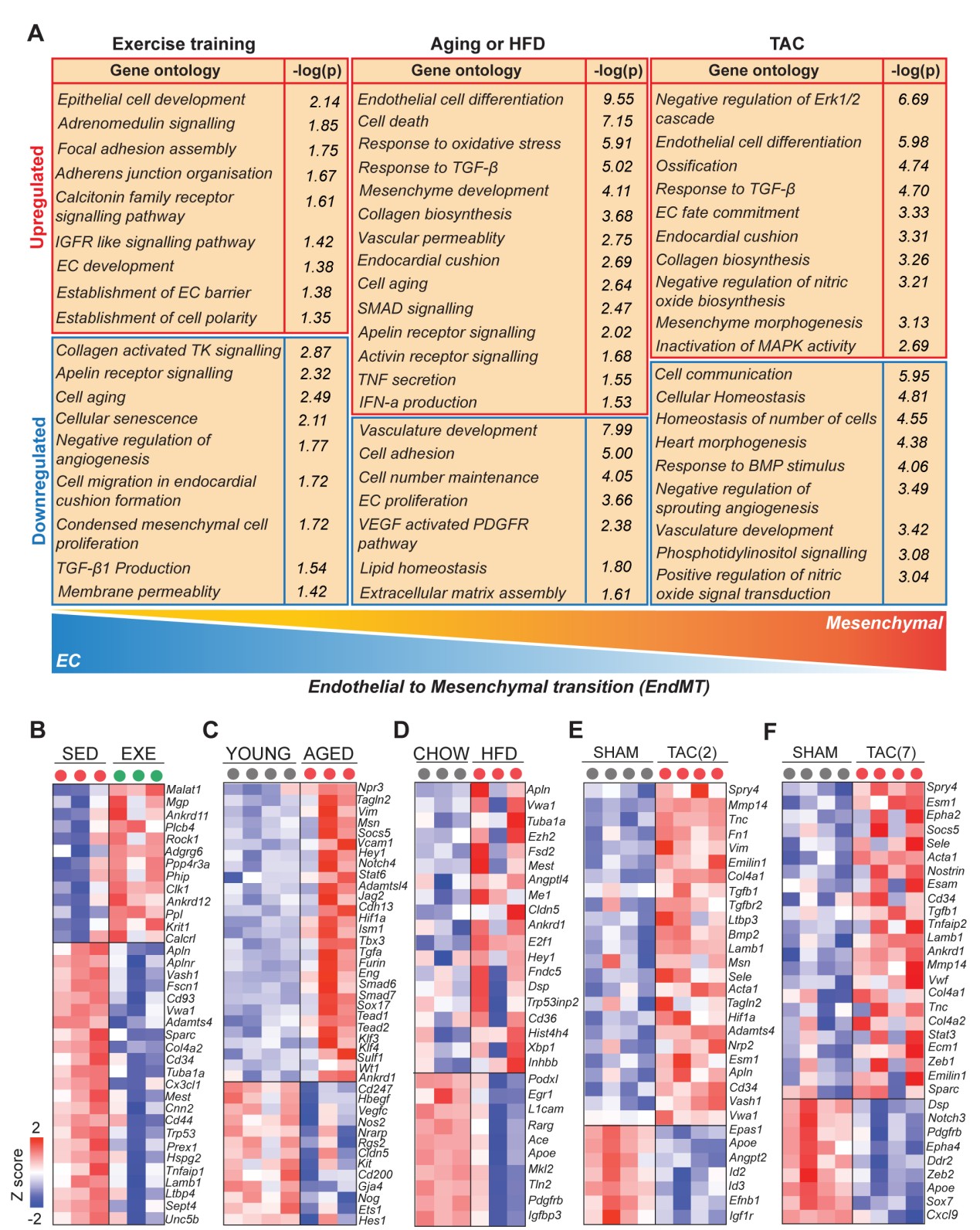

**Figure 3.** Cardiovascular disease (CVD) risk factors activate mesenchymal gene expression in cardiac endothelial cells (ECs). (**A**) Gene ontology analysis of the up- and downregulated genes. Note the opposite changes induced by exercise training compared to the CVD risk factors. (**B–F**) Heatmaps showing the differential gene expression of endothelial and mesenchymal genes previously associated with endothelial-to-mesenchymal transition (EndMT). Genes are selected based on published data sets (references are found in *Figure 3—source data 1*). In all panels, the up- and

*Figure 3 continued on next page*

*Figure 3 continued*

downregulated genes with the false discovery rate (FDR; Benjamini–Hochberg adjusted p-value) threshold of 0.05 were considered. In the heatmap, each color-coded circle (red, green, and black) indicates an individual biological sample within each experimental group. N = 3–4 male mice/group.

The online version of this article includes the following source data and figure supplement(s) for figure 3:

**Source data 1.** Genes and reference list for endothelial and mesenchymal genes indicated in the *Figure 3B–F* heat map.
**Source data 2.** Source data for *Figure 3B, C, D, E and F*.
**Figure supplement 1.** Cardiovascular disease (CVD) risk factors induce inflammatory gene expression in cardiac endothelial cells (ECs).
**Figure supplement 1—source data 1.** Source data for *Figure 3—figure supplement 1A, B, C, D and E*.
**Figure supplement 2.** Obesity and pressure overload induce SASP gene expression and senescence in the heart.
**Figure supplement 2—source data 1.** Source data for *Figure 3—figure supplement 2B, C, D and E*.
**Figure supplement 3.** QPCR validation of selected genes in the cardiac endothelial cells (ECs) of aged, obese, and exercise trained mice.
**Figure supplement 3—source data 1.** Source data for *Figure 3—figure supplement 3A, B, C, D, E and F*.

mesenchymal markers, indicating possible reversal of the process (*Figure 3E,F*). Strikingly, exercise training downregulated several EndMT genes (*Fscn1, Cd93, Vwa1, Sparc, Tuba1a, Cd44, Trp53, Col4a2, Mest, Cnn2, Tnfaip1, Lamb1, Ltbp4,* and *Unc5b*), the angiogenesis inhibitor gene *Vash1,* and the endothelial activation marker *Apln* and its receptor *Aplnr,* whereas it upregulated the expression of *Malat1, Mgp, Krit1, and Calcrl* (*Figure 3B*). We validated the results using an expanded set of samples by qPCR for *Apln, Vim, Tgfbr2, Vash1, Sparc,* and *Tgfb1* (*Figure 3—figure supplement 3A–F*).

## *Serpinh1* expression is increased by aging and obesity and repressed by exercise training

To identify genes, which could mediate the negative effect of aging and obesity and the protective effects of exercise, we performed gene overlap analysis of DEGs from these three experimental interventions. We found four genes significantly affected by all treatments, of which two genes (*Serpinh1* and *Vwa1*) were upregulated by aging and HFD and downregulated by exercise training. The other two genes (*Mest* and *Fhl3*) were upregulated by HFD and downregulated by exercise training and aging (*Figure 4A–C*). We performed an in silico secretome analysis to characterize the properties of the identified genes using MetaSecKB database (*Figure 4D*). Both *Serpinh1* and *Vwa1* contain a signal peptide for secretion, indicating they could act as angiocrines in autocrine and/or paracrine fashion.

We focused on *Serpinh1,* as it has a known role as a collagen chaperone and has been linked to fibrosis (*Ito and Nagata, 2019*), making it an attractive candidate. We validated the endothelial *Serpinh1* expression by qPCR (*Figure 4C*), and at single cell level using Tabula Muris database (*Tabula Muris Consortium et al., 2018*) and cardiac EC atlas from the Carmeliet lab (*Kalucka et al., 2020*). The scRNAseq analysis revealed that *Serpinh1* is expressed in variety of cell types within the mouse heart, including fibroblasts, myofibroblasts, smooth muscle cells, ECs, endocardial cells, and to lesser extent in cardiomyocytes (*Figure 4—figure supplement 1A–D*). In ECs, *Serpinh1* was found to be expressed throughout all EC clusters, with the highest expression in the apelin-high cluster marking activated ECs (*Figure 4—figure supplement 2A–F*). Interestingly, the expression of mesenchymal markers such as *Tagln2, Vim,* and *Smtn* was also high in this cluster. Next, we analyzed the expression of SERPINH1 (also called as HSP47) in healthy human heart and in human cardiac ECs. Immunohistochemistry demonstrated SERPINH1 to be highly expressed throughout the coronary vasculature and in fibroblasts in human heart, and weak staining was also detected in cardiomyocytes (*Figure 4E–G, Figure 4—figure supplement 1D*). Analysis using the EndoDB database (E-GEOD-43475) showed that the expression of *SERPINH1* is highly similar in both veins and arteries and in different tissues (heart, lungs, liver, human cardiac arterial EC (HCAEC), and human umbilical venous EC [HUVEC]) (*Figure 4—figure supplement 3A*). In human cardiac ECs, SERPINH1 was localized perinuclearly, similar to what has been demonstrated in other cells types, and consistent with the ER retention motif in its N-terminus (*Figure 4E*; *Masuda et al., 1994*; *Razzaque et al., 1998*; *Honzawa et al., 2014*).

We next tested, if exercise training can attenuate the expression of *Serpinh1, Vwa1,* and selected markers of TGF-β signaling/EndMT also in aged mice. Of the studied genes, mRNA expression of

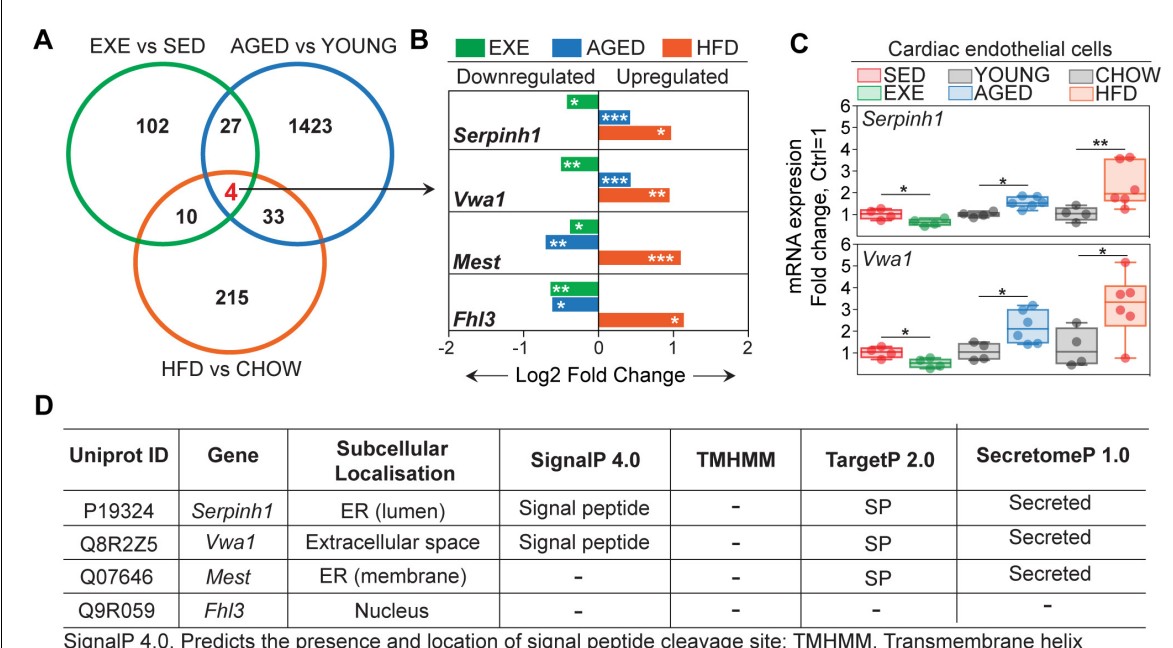

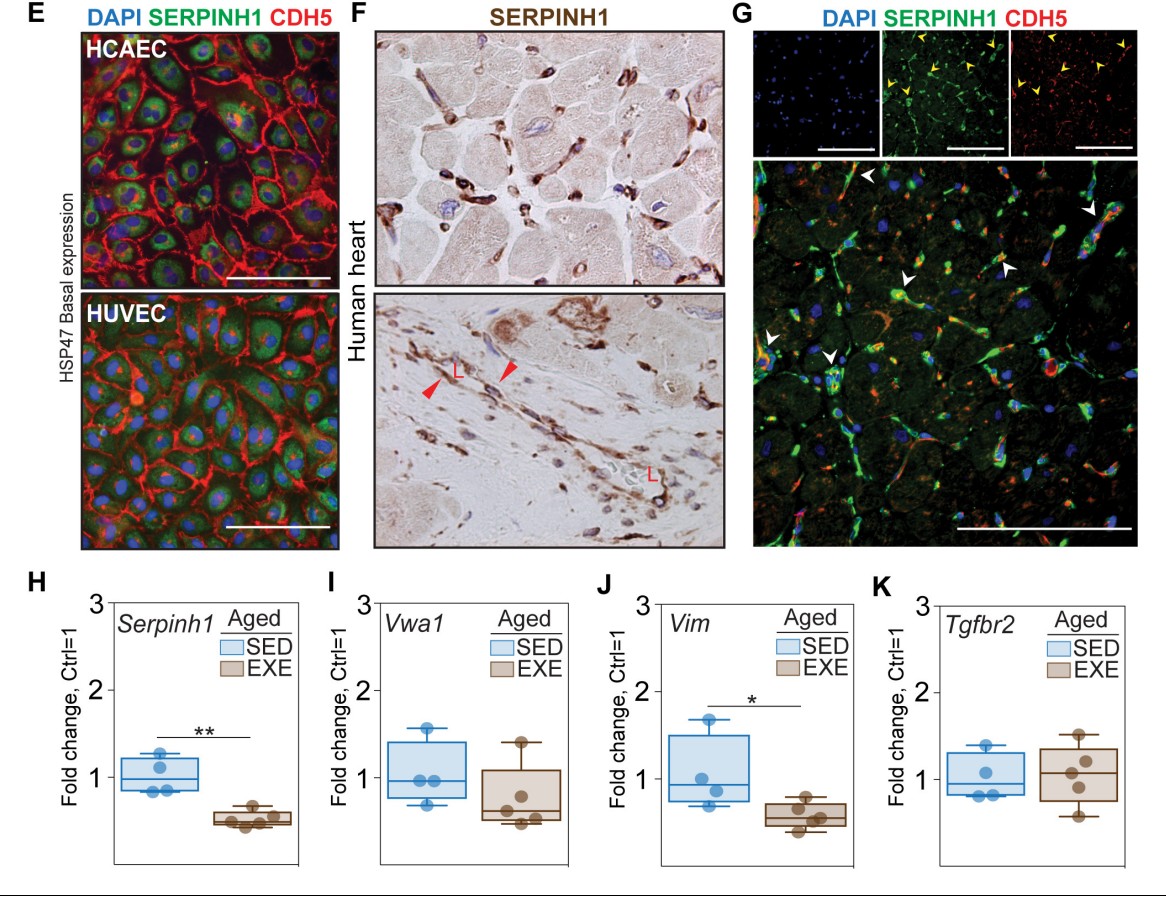

**Figure 4.** *Serpinh1* expression is increased by aging and obesity and repressed by exercise training. (**A**) A Venn diagram showing the overlap of differentially expressed genes between the experiments. Four genes were identified to be significantly affected by aging, obesity, and exercise (*Serpinh1, Vwa1, Mest,* and *Fhl3*). (**B**) Bar plot showing the expression pattern of these four genes. In panels (**A** and **B**), the up- and downregulated genes with the false discovery rate (FDR; Benjamini–Hochberg adjusted p-value) threshold of 0.05 were considered to be significant (N = 3–4 male

*Figure 4 continued on next page*

*Figure 4 continued*

mice/group). (C) qPCR validation of *Serpinh1* and *Vwa1* normalized to *Hprt1* (N = 4–6 male mice/group). (D) In silico secretome analysis of the identified genes. (E–G) Representative immunofluorescent and immunohistochemistry images showing the expression of SERPINH1 in human endothelial cell (EC) and human heart samples. Red arrowhead in the bottom panel F indicates the expression in large vessels and 'L' indicates vessel lumen. White arrowheads in the panel G denote the co-expression of SERPINH1 and CDH5 in coronary vessels (yellow signal). (H–K) mRNA expression of *Serpinh1, Vwa1, Vim,* and *Tgfbr2* in the cardiac ECs of sedentary and exercise trained aged mice (N = 4–5 female mice/group). Scale bar 100 μm. Data is presented as mean ± SEM. Student's t-test was used, *p<0.05, **p<0.01, ***p<0.001.

The online version of this article includes the following source data and figure supplement(s) for figure 4:

**Source data 1.** Source data for *Figure 4B, C, H, I, J and F*.
**Figure supplement 1.** Expression of *Serpinh1* in different cardiac cell types and in the human heart.
**Figure supplement 2.** *Serpinh1* expression in different subsets of cardiac endothelial cell (EC).
**Figure supplement 3.** SERPINH1 RNA levels in the human arterial and venous endothelial cells.

*Serpinh1* and *Vim* were significantly repressed by exercise training, and there was a tendency also for *Vwa1* (*Figure 4H–K*).

## Overexpression of SERPINH1 induces mesenchymal features in human ECs

To study the effects of SERPINH1 in human ECs, we produced lentiviral vector encoding myc-tagged hSERPINH1. Both HUVECs and HCAECs were analyzed. SERPINH1 protein was localized similar to the native protein (*Figure 5B*), and the expression was verified by western blotting (*Figure 5—figure supplement 1A*). Overexpression of SERPINH1 altered the cellular morphology characterized by impaired or discontinuous vascular endothelial cadherin junctions, increased stress fiber formation, and larger cell size (*Figure 5A,B*). Furthermore, analysis of EC and mesenchymal cell-related transcripts demonstrated significant repression of EC markers (*CD31, CDH5, TIE1, NRARP,* and *ID1*) and induction of a proliferation gene CCND1, and mesenchymal/EndMT markers (*TAGLN, aSMA, CD44, VIM, NOTCH3, ZEB2, SLUG, FN1, VCAM1,* and *ICAM1*) (*Figure 5C*). VE-cadherin downregulation was also confirmed at protein level (*Figure 5D*) and increased aSMA expression by immunofluorescence staining (*Figure 5E*). We also analyzed the effect of SERPINH1 on cellular senescence. SA-β-galactosidase staining showed increased number of cells undergoing senescence and there was a clear upregulation of senescence-associated genes (*Figure 5G,H*).

Transcriptomic changes pointed toward activated TGF-β signaling and oxidative stress in response to all of the CVD risk factors. Both are known to contribute to EC dysfunction and EndMT, and thus we tested if they act as upstream regulators of SERPINH1. Indeed, our results show that TGF-β1-treatment of HCAECs significantly upregulated the expression of *SERPINH1* together with other known EndMT markers, and there was also small but significant induction of *SERPINH1* by hydrogen peroxide treatment (*Figure 5F*).

## SERPINH1 is needed for collagen 1 deposition by ECs

To investigate the significance of SERPINH1 depletion in human cardiac ECs, HCAECs were transduced with four independent shSERPINH1 lentiviral constructs. The constructs induced approximately 80% deletion of SERPINH1 mRNA (*Figure 6D*). The cell morphology was not affected after 2 days (*Figure 6A*), but 10 days of silencing significantly changed EC morphology and decreased the cell density in culture (*Figure 6B*), suggesting that SERPINH1 might play a role in EC homeostasis and survival. SERPINH1 silencing significantly inhibited collagen fibril deposition, detected by immunohistochemistry for type 1 collagen (*Figure 6B,C*). Only the cells transduced with the construct #1 could produce some extracellular collagen 1, and these cells also survived better than the cells transduced with constructs #2, #3, or #4 (*Figure 6B,C*). Next, we treated the cells with TGF-β1 and hydrogen peroxide for 5 days to induce EndMT features, as described previously (*Evrard et al., 2016*; *Magenta et al., 2011*). We used the shSERPINH1 (#1) construct, because from the other silencing constructs not enough cells survived for the experiments. The results indicated that silencing of SERPINH1 prevented the appearance of Taglin-positive cells, a commonly used readout for EndMT, which were observed in the control cells (*Figure 6E*).

We also studied the effect of SERPINH1 on cell proliferation/migration. In the scratch wound assay, overexpression of SERPINH1 significantly promoted wound closure (*Figure 7A,B*), whereas

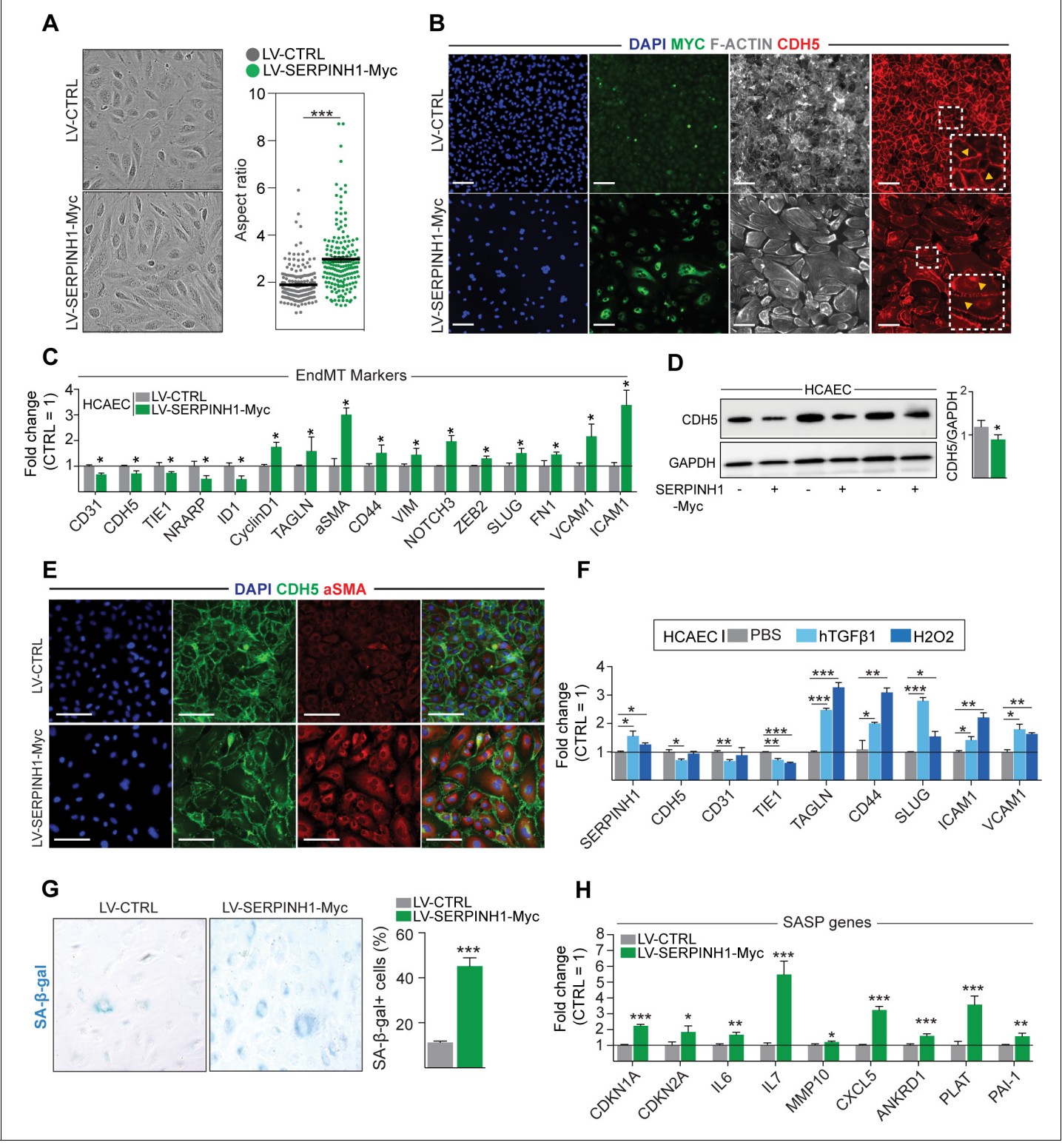

**Figure 5.** Overexpression of SERPINH1 modifies the endothelial cell (EC) phenotype and induces mesenchymal gene expression in human cardiac ECs. (A) Representative phase-contrast images of live human cardiac arterial EC (HCAEC) transduced with LV-CTRL and LV-SERPINH1-Myc, and quantification of the aspect ratio (length to width ratio) of the cell. (B) Representative immunofluorescent images showing the expression of Myc-tagged SERPINH1 in green, F-Actin in gray, and CDH5/VE-Cadherin in red. The inset within the white box shows magnified view of VE-Cadherin junctions in HCAECs. (C) qPCR analysis of endothelial and mesenchymal markers in SERPINH1 overexpressing cells. (D) Western blot analysis and quantification of CDH5/VE-cadherin expression in the SERPINH1 overexpressing HCAECs (normalized to GAPDH). (E) Representative

*Figure 5 continued on next page*

*Figure 5 continued*

immunofluorescent images showing DAPI in blue, CDH5/VE-Cadherin in green, and α-smooth muscle actin (aSMA) in red. (F) qPCR analysis of SERPINH1 and EndMT markers in HCAECs stimulated with TGF-β1 (50 ng/ml) or $H_2O_2$ for 5 days. (G) Representative images and quantification of SA-β-gal+ senescent cells (in blue) normalized to total nuclei (%) in SERPINH1 overexpressing and control cells. (H) qPCR analysis of senescence-associated secretory phenotype (SASP) genes in HCAECs transduced with LV-CTRL and LV-SERPINH1-Myc. In panels A, C, D, F, G, and H, N = 3 biological replicates/group were analyzed. Scale bar 100 µm. Data is presented as mean ± SEM. Student's t-test was used, *p<0.05, **p<0.01, ***p<0.001.

The online version of this article includes the following source data and figure supplement(s) for figure 5:

**Source data 1.** Source data for *Figure 5A, C, D, F, G and H*.
**Source data 2.** Source data for *Figure 5D*.
**Figure supplement 1.** SERPINH1 overexpression in endothelial cells.
**Figure supplement 1—source data 1.** Source data for *Figure 5—figure supplement 1A*.

silencing of SERPINH1 for 2 days significantly decreased EC proliferation/migration (*Figure 7C,D*). Cell proliferation was slightly increased by SERPINH1 overexpression, whereas silencing almost completely blocked proliferation, as determined by EdU incorporation (*Figure 7E–G*).

## Discussion

Here we have used transcriptomic profiling to decipher how the major CVD risk factors aging, obesity, and pressure overload remodel cardiac ECs, and how the protective effects of exercise are mediated. The results demonstrate that the CVD risk factors activate transcriptional programs promoting cell aging, senescence, TGF-β activation, inflammation, and oxidative stress in cardiac ECs. Importantly, exercise attenuated these same pathways, even in healthy mice. Furthermore, we found that aging, obesity, and pressure overload induced mesenchymal gene programs in cardiac ECs, which can contribute to dysfunctional endothelium and CVD development. Analysis of potential disease-promoting genes identified *Serpinh1* to be induced by aging and obesity, while its expression was significantly repressed by exercise, also in old mice. Mechanistically, SERPINH1 was induced by TGF-β and ROS, and the overexpression of SERPINH1 increased cell size and stress fiber formation, weakened cell–cell junctions, and promoted mesenchymal and senescence-associated gene expression in human cardiac ECs. Immunohistochemistry of human hearts showed that SERPINH1 is abundantly expressed throughout the cardiac vasculature.

The largest dysregulation of the cardiac EC transcriptome was found in aged mice, followed by obesity and pressure overload. Exercise training affected a smaller number of transcripts, which can be accounted, at least partly, to the young and healthy control mice, which could move unrestrictedly in their home cages. Interestingly, however, most of the pathways activated by CVD risk factors were the same that were repressed by exercise training, highlighting the potential of physical activity to improve cardiovascular health via modulating EC phenotype and function. The positive effects of exercise on skeletal muscle and cardiac angiogenesis have been described previously (*Hudlicka et al., 1992*), but exercise-induced molecular changes in ECs have not been characterized. It is important to note that here we studied the chronic adaptations to exercise training, and not the acute responses, which likely explains why more genes were found to be downregulated than upregulated in these mice. The effects of exercise training in cardiac ECs were associated with EC homeostasis and stabilization, with increased expression of genes related to establishment of EC barrier, polarity, and focal adhesion. Importantly, exercise induced repression of inflammatory, permeability, senescence, and mesenchymal gene networks. It also attenuated the expression of apelin, which is considered as a marker of activated ECs, and also its receptor Aplnr. This suggests that regular exercise training promotes stabilization and quiescence in cardiac ECs and prevents cellular aging. Aging and obesity, on the other hand, are known to contribute to capillary rarefaction and/or dysfunction (*Cines et al., 1998*; *Gimbrone and García-Cardeña, 2016*; *López-Otín et al., 2013*; *Ungvari et al., 2010*), and another novel aspect in this study was the comparison of several CVD risk factors to identify common pathways and genes, which could drive the pathogenesis in cardiac disease, and could be considered as potential therapeutic targets. ECs would provide an attractive target for drug development, as they are the first cells to encounter drugs in the bloodstream.

Dysfunctional endothelium likely contributes to more diseases than any other tissue in the body as it affects all organs. On the other hand, endothelium could act as an important mediator of the

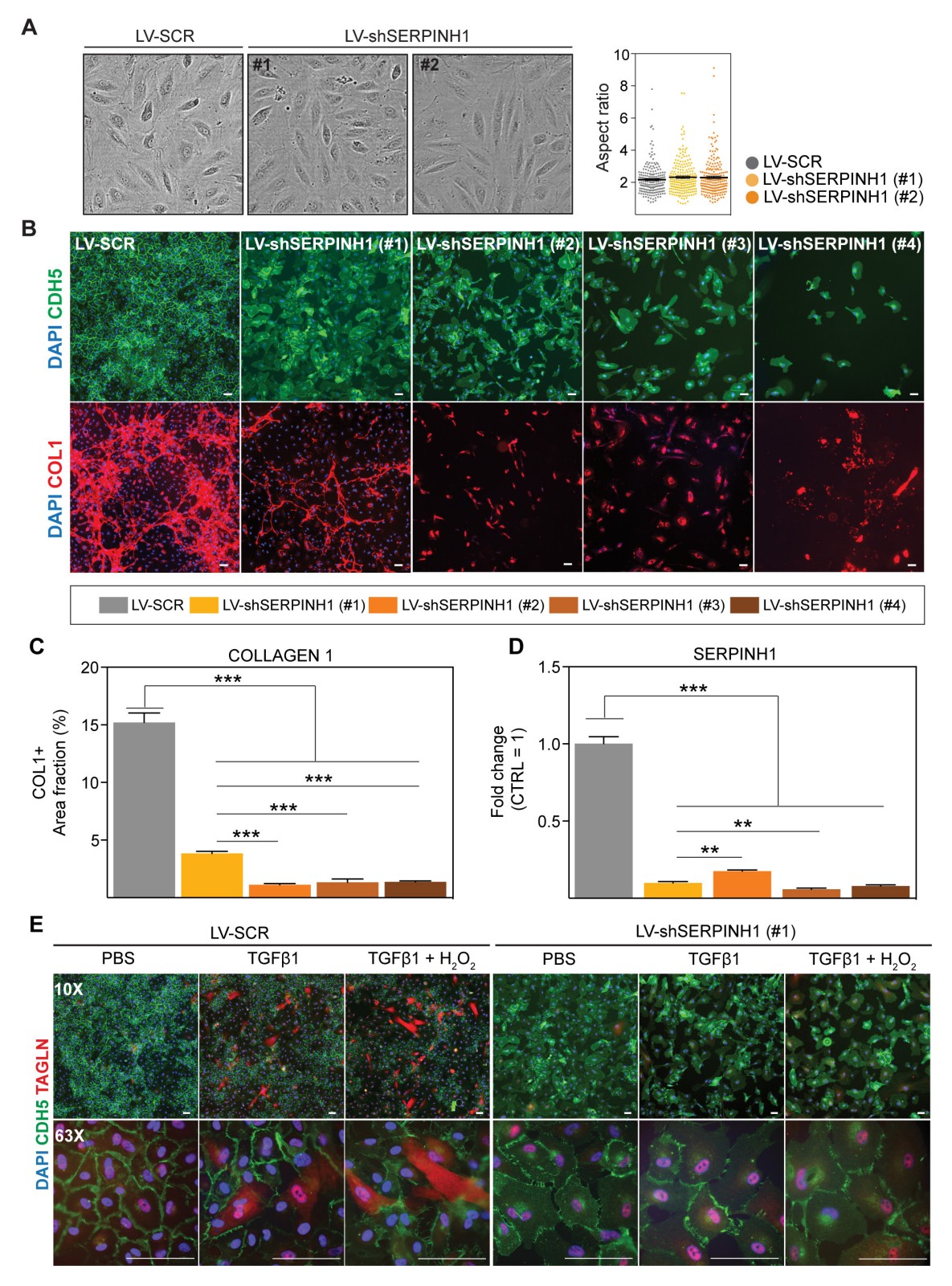

**Figure 6.** SERPINH1 silencing in human cardiac endothelial cell (EC) inhibits collagen production and EndMT. (**A**) Representative phase contrast images of live human cardiac arterial ECs (HCAECs) transduced with LV-SCR and LV-shSERPINH1 (#1 and #2) and quantification of the aspect ratio (length to width ratio) of the cells 48 hr after transduction. (**B**) Representative CDH5/VE-Cadherin immunofluorescent images (green) showing the cell morphology and density after 10 days of SERPINH1 silencing. Collagen 1 staining is shown in red, and quantification of collagen 1 is shown in

*Figure 6 continued on next page*

*Figure 6 continued*

C. (D) qPCR analysis of SERPINH1 deletion levels using four independent constructs. (E) Representative immunofluorescent images showing TAGLN expression in the control and SERPINH1 silenced HCAECs treated with recombinant human TGF-β1 with and without $H_2O_2$ for 5 days. In the panels (A, C and D), N = 3 biological replicates/group were analyzed. Scale bar 100 μm. Data is presented as mean ± SEM. Student's t-test was used, *p<0.05, **p<0.01, ***p<0.001.

The online version of this article includes the following source data for figure 6:

**Source data 1.** Source data for *Figure 6A, C and D*.

health-promoting effects of exercise in a variety of tissues. Our finding that aging, obesity, and pressure overload induce mesenchymal gene programs in cardiac ECs adds to the increasing evidence that activated endothelial TGF-β signaling and acquisition of mesenchymal features play an important role in the development of EC dysfunction and cardiac diseases (*Kovacic et al., 2019*; *Chen et al., 2015*; *Chen et al., 2019*; *Xiong et al., 2018*). Importantly, genes related to TGF-β production and cellular aging were repressed by exercise, highlighting the potential of exercise training in preventing and delaying the development of CVD. The molecular mechanisms of exercise-mediated repression of TGF-β signaling are not known. Nitric oxide (NO) has been previously shown to attenuate TGF-β/SMAD2 signaling in ECs, whereas mice lacking endothelial NO synthase activity presented increased TGF-β signaling and collagen 1 in their aortas (*Saura et al., 2005*). Increased blood flow during exercise induces eNOS expression and NO production, which could repress TGF-β activity in the vasculature.

Reduced TGF-β activation was also recently reported in whole heart lysates in exercised rats (*Lin et al., 2020*), and enhanced TGF-β signaling was also suggested to be a negative regulator of exercise response in human skeletal muscle (*Böhm et al., 2016*).

The activation of TGF-β signaling pathway has been implicated as a driving force for EndMT (*Evrard et al., 2016*; *Cooley et al., 2014*; *James and Rafii, 2014*; *Bischoff, 2019*; *Bischoff et al., 2016*). Several studies have recently suggested that EndMT could contribute to the development of various CVDs (*Zeisberg et al., 2007*; *Kovacic et al., 2019*; *Li et al., 2018*; *Sánchez-Duffhues et al., 2018*), but currently there is a lack of understanding of the causal relationships and mechanisms linking EndMT and CVD (*Kovacic et al., 2019*). Furthermore, whether the transition from ECs to mesenchymal cells occurs completely in various CVDs is still actively debated in the literature. It has been suggested that pathological EC activation will result in acquired EndMT features for example expression of mesenchymal genes, without full transformation from one cell type to another (*Chen et al., 2020*). This is in line with our findings, as only cells with high CD31 expression and with no expression of CD45, CD140a, and Ter119 were included in our analyses. Thus, all the analyzed cells were ECs, but in the CVD risk factor groups they demonstrated increased mesenchymal marker expression. Long-term lineage tracing of ECs in response to CVD risk factors would provide further knowledge if and to what extent full transformation of ECs to mesenchymal cells occurs in cardiac vasculature. Our results, however, demonstrate that ECs acquire mesenchymal features due to CVD risk factors, which likely results in EC dysfunction even without full EndMT.

It was not surprising that SASP-genes were induced in old mouse ECs, but the observation that this was also seen in the obese and pressure overloaded mice caught our attention. In ECs of obese mice, increased *p53* expression has been reported, which led to reduced eNOS phosphorylation both in vitro and in vivo (*Yokoyama et al., 2014*). Combined with our data, this could then result in increased TGF-β activity (*Saura et al., 2005*), linking senescence and TGF-β signaling. We also observed significantly more SA-β-galactosidase positive cells in the hearts of obese mice compared to the lean mice. These cells were often found in clusters, and in addition to ECs, they could also be other non-myocytes. Thus the significance of these cells to cardiac vasculature remains to be further studied. Endothelial deletion of *p53* has also been demonstrated to protect against pressure overload-induced cardiac dysfunction and fibrosis, suggesting that increased p53 and other senescence-associated genes are important mediators of EC dysfunction (*Gogiraju et al., 2015*).

To identify possible pathology-driving genes, which would be common for several risk factors, we performed gene overlap analysis using all data sets. Two genes, *Serpinh1* and *Vwa1*, were found to be significantly increased by both aging and obesity and decreased by exercise, suggesting that they could act as common mediators of EC dysfunction. We focused in this study on *Serpinh1*, as it is a collagen chaperone and has been shown to contribute to tissue fibrosis (*Ito and Nagata, 2019*;

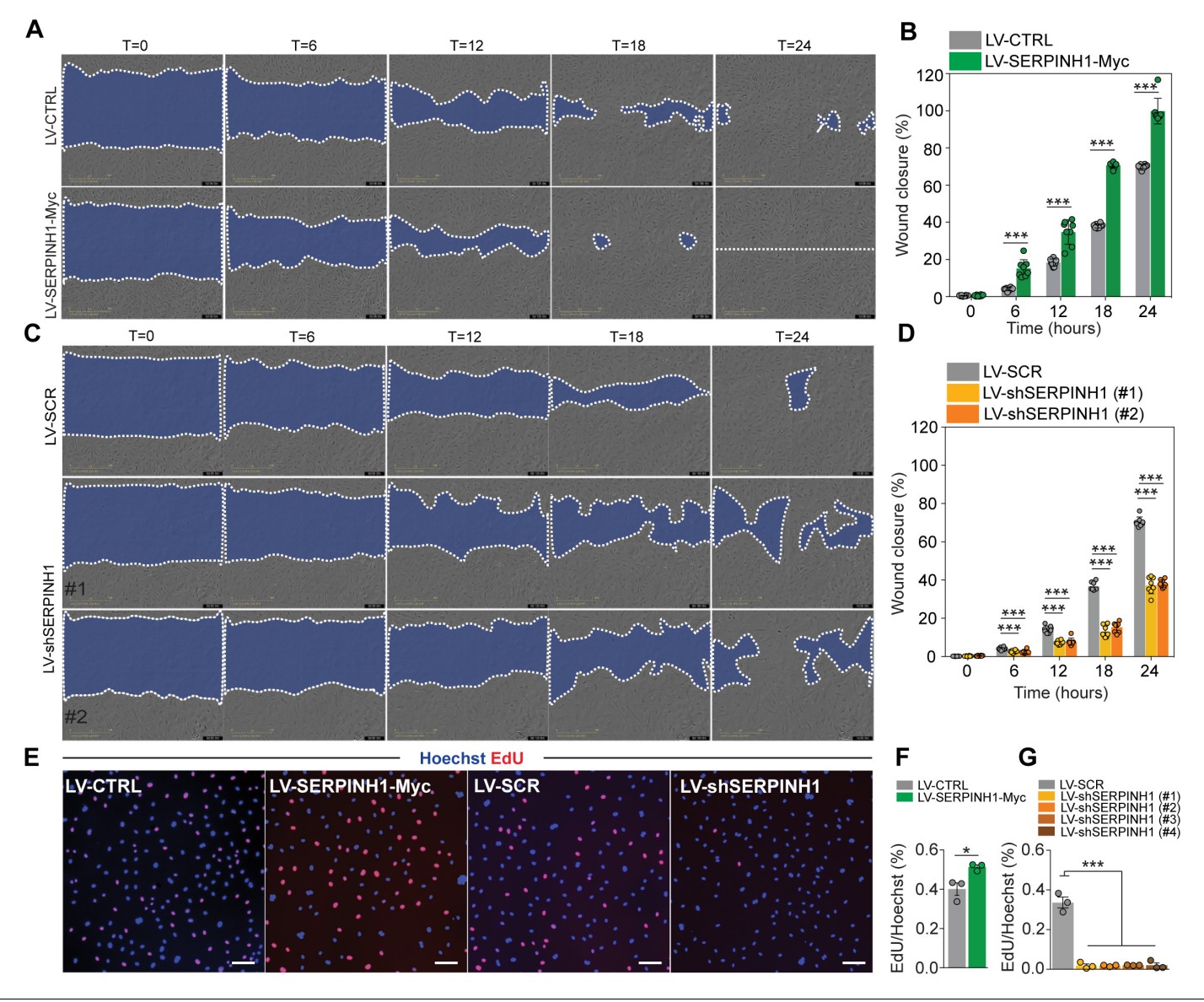

**Figure 7.** SERPINH1 overexpression enhances and silencing inhibits wound closure in vitro. (**A and B**) Representative phase contrast images of scratch wound healing assay performed in human cardiac arterial ECs (HCAECs) treated with LV-CTRL and LV-SERPINH1, and quantification of the wound closure (%) with respect to time (hours). (**C and D**) Representative phase contrast images of scratch wound healing assay performed in HCAECs treated with LV-SCR and LV-shSERPINH1 (#1 and #2), and quantification of the wound closure (%) with respect to time (hours). (**E–G**) Representative immunofluorescent images of EdU incorporation in HCAECs treated with LV-Ctrl, LV-SERPINH1-Myc, LV-Scr, and LV-shSERPINH1 (#1, #2, #3, and #4), and quantification of EdU+ nuclei (red) normalized to Hoechst+ nuclei (blue). In the panels (**A and C**), the blue area within the white dotted region indicates the wound area. In the panels (**B and D**), N = 8 biological replicates/group and in (**F and G**), N = 3 biological replicates/group were analyzed. Scale bar 100 µm. Data is presented as mean ± SEM. Student's t-test was used, *p<0.05, **p<0.01, ***p<0.001.

The online version of this article includes the following source data for figure 7:

**Source data 1.** Source data for *Figure 7B, D, F and G*.

*Khalil et al., 2019*), an important feature of many cardiac diseases. Recently, it was demonstrated using *Serpinh1* cell type-specific knockout mice that *Serpinh1/Hsp47* in myofibroblasts is an important regulator of pathologic cardiac fibrosis (*Khalil et al., 2019*). In line with our results, collagen 1 production was decreased also in the EC-specific *Serpinh1* deficient hearts in TAC-operated mice (*Khalil et al., 2019*). In human cardiac ECs, our results placed SERPINH1 downstream of TGF-β and ROS, and demonstrated that its overexpression promoted mesenchymal features and senescence .

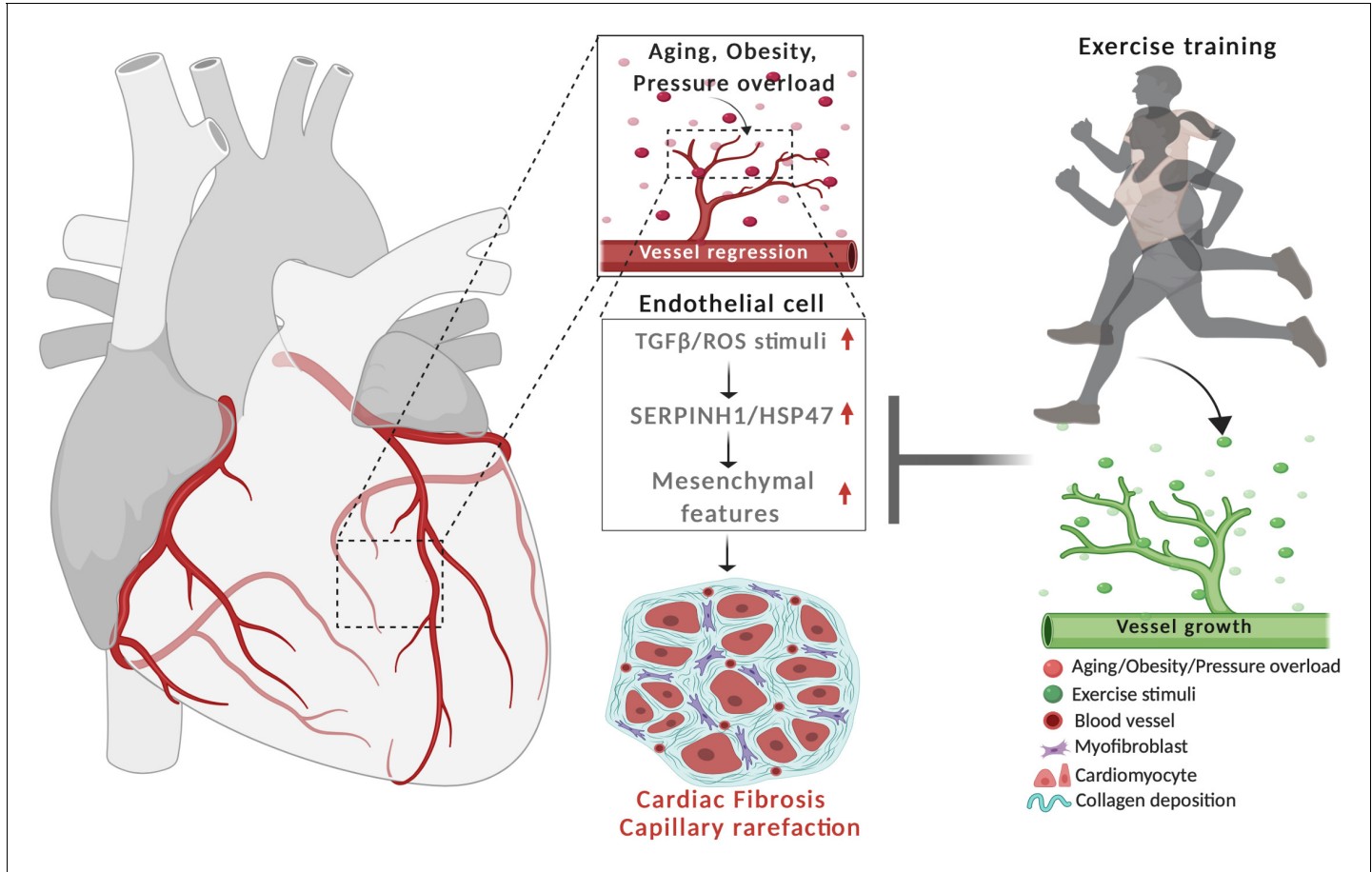

**Figure 8.** Schematic demonstrating the cardiovascular disease risk factor mediated activation of TGF-β signaling and acquisition of mesenchymal gene features in cardiac EC. CVD risk factors aging, obesity and pressure overload trigger the regression of coronary vasculature by activating TGF-β/ROS signaling pathways and cellular senescence. These induce the expression of SerpinH1/HSP47 and mesenchymal gene signature. SerpinH1/HSP47 and EndMT are both involved in the development of tissue fibrosis by increasing collagen deposition in the extracellular matrix. Exercise training, in turn, increases coronary vasculature density, EC number, and represses TGF-β signaling, mesenchymal gene expression, and cellular aging related pathways.

Furthermore, SERPINH1 was found to be important for extracellular collagen 1 deposition and EC proliferation and migration. Overexpression of SERPINH1 slightly increased proliferation, but the effect was more pronounced on migration, whereas silencing of SERPINH1 almost completely blocked proliferation. Silencing also prevented the TGF-β induced appearance of TAGLIN-positive cells in human cardiac EC, which is considered as a marker for EndMT (*Evrard et al., 2016*; *Magenta et al., 2011*). It is counterintuitive that SERPINH1 increased senescence markers that inhibit cell proliferation, but at the same time increased proliferation and migration. Proliferation was evaluated 48 hr after transduction, thus it is possible that at this time point, the induced mesenchymal properties override the senescence signals, which might take over at later time points, reflected as increased cell size and SA-beta galactosidase staining.

Based on the publicly available single-cell RNA sequencing data and immunohistochemistry of the human heart samples, SERPINH1 is abundantly expressed in all cardiac endothelial populations, as well as in arterial and venous ECs in other tissues. Not much is known about the role of SERPINH1 in heart disease. In a study by Kato et al., SERPINH1 was found to colocalize with several other EndMT markers in some of the ECs in left atrium in patients with atrial fibrillation, a disease which is often related to fibrosis (*Kato et al., 2017*). To further the translational potential, the role of endothelial SERPINH1 in aged, obese, and hypertensive human hearts needs to be determined.

In the in vivo experiments for endothelial RNA sequencing analyses, we have used male mice, which were age and gender matched in exercise, obesity, and TAC models. Thus, most of the results

represent responses in male mice, whereas in the exercise training experiment in old mice, female mice were used. Some of the responses, for example the repression of *Serpinh1* in exercised mice, were similar to those observed in young male mice; however, some of the changes were not significant in old female mice. It will be important to determine if the findings presented here in male mice are also valid in females, and even more interestingly, in humans.

In conclusion, our data demonstrate that the major CVD risk factors significantly remodel the cardiac EC transcriptome promoting cell senescence, oxidative stress, TGF-β signaling, and mesenchymal gene features, whereas exercise training provided opposite and protective effects (*Figure 8*). SERPINH1 was identified as one of the downstream effectors of TGF-β, which could provide a novel therapeutic target in ECs.

# Materials and methods

### Key resources table

| Reagent type (species) or resource | Designation | Source or reference | Identifiers | Additional information |
|---|---|---|---|---|
| Genetic reagent (*M. musculus*) | C57BL/6J | Janvier Labs | RRID:IMSR_JAX:000664 | |
| Cell line (*Homo sapiens*) | HCAEC, HUVEC | PromoCell | | |
| Transfected construct (*Homo sapiens*) | SERPINH1 (*Homo sapiens*), shRNA(#1) | TRC library database, Broad institute | TRCN0000003590 | Lentiviral construct to transfect and express the shRNA |
| Transfected construct (*Homo sapiens*) | SERPINH1 (*Homo sapiens*), shRNA(#2) | TRC library database, Broad institute | TRCN0000003594 | Lentiviral construct to transfect and express the shRNA |
| Transfected construct (*Homo sapiens*) | SERPINH1 (*Homo sapiens*), shRNA(#3) | TRC library database, Broad institute | TRCN0000003593 | Lentiviral construct to transfect and express the shRNA |
| Transfected construct (*Homo sapiens*) | SERPINH1 (*Homo sapiens*), shRNA(#4) | TRC library database, Broad institute | TRCN0000003591 | Lentiviral construct to transfect and express the shRNA |
| Transfected construct (*Homo sapiens*) | FUW-SERPIH1-Myc (*Homo sapiens*) | This paper | | Lentiviral construct to transfect and express overexpress SERPINH1-Myc |
| Biological sample (*Homo sapiens*) | Heart samples from donors | Helsinki University Hospital | | |
| Antibody | (Rat monoclonal), FITC-CD31 | Invitrogen | RM5201, RRID:AB_10373983 | FACS (1:100) |
| Antibody | (Rat monoclonal), Pacificblue-CD45 | Biolegend | 103125, RRID:AB_493536 | FACS (1:100) |
| Antibody | (Rat monoclonal), Pacificblue-Ter119 | Biolegend | 116231, RRID:AB_2149212 | FACS (1:100) |
| Antibody | (Rat monoclonal), PE-Cyanine7-CD140a | eBioscience | 25-1401, RRID:AB_2573399 | FACS (1:100) |
| Antibody | (Rat monoclonal), CD16/CD32 (Fc blocker) | BD Biosciences | 553142, RRID:AB_394657 | FACS (1:100) |
| Antibody | (Rat monoclonal), CD31 | BD Pharmingen | 553370, RRID:AB_394816 | Immunofluorescent (1:500) |
| Antibody | (Rabbit monoclonal), VEcadherin | Cell Signaling | 2500S, RRID:AB_10839118 | Immunofluorescent or western blotting (1:500) |
| Antibody | (Sheep Polyclonal), Tagln | R&D Biosystems | AF7886 | Immunofluorescent (1:500) |

*Continued on next page*

*Continued*

| Reagent type (species) or resource | Designation | Source or reference | Identifiers | Additional information |
|---|---|---|---|---|
| Antibody | (Mouse Monoclonal), c-MYC | Thermo Fisher | 13-2500, RRID:AB_2533008 | Immunofluorescent or western blotting (1:500) |
| Antibody | (Mouse Monoclonal), HSP47/SERPINH1 | Enzo Life Sciences | ADI-SPA-470-D, RRID:AB_2039239 | Immunofluorescent immunohistochemistry or western blotting (1:1000) |
| Antibody | (Rabbit Polyclonal), Collagen 1 | Abcam | ab34710, RRID:AB_731684 | Immunofluorescent (1:1000) |
| Antibody | (Mouse Monoclonal), aSMA | Sigma-Aldrich | A5228, RRID:AB_262054 | Immunofluorescent (1:500) |
| Antibody | (Mouse Monoclonal), GAPDH | Millipore | CB1001, RRID:AB_2107426 | Western blotting (1:500) |
| Sequence-based reagent | hSERPINH1_F | This paper | SYBR green PCR primers | ATGAGAAATTCCACCACAAGATG |
| Sequence-based reagent | hSERPINH1_R | This paper | SYBR green PCR primers | GATCTTCAGCTGCTCTTTGGTTA |
| Sequence-based reagent | hCD31_F | This paper | SYBR green PCR primers | CTGCTGACCCTTCTGCTCTGTTC |
| Sequence-based reagent | hCD31_R | This paper | SYBR green PCR primers | GGCAGGCTCTTCATGTCAACACT |
| Sequence-based reagent | hCDH5_F | This paper | SYBR green PCR primers | CGTGAGCATCCAGGCAGTGGTAGC |
| Sequence-based reagent | hCDH5_R | This paper | SYBR green PCR primers | GAGCCGCCGCCGCAGGAAG |
| Sequence-based reagent | hTIE1_F | This paper | SYBR green PCR primers | ACCCGCTGTGAACAGGCCTGCAGAGA |
| Sequence-based reagent | hTIE1_R | This paper | SYBR green PCR primers | CTTGGCACTGGCTTCCTCT |
| Sequence-based reagent | hCYCLIND1_F | This paper | SYBR green PCR primers | GCGGAGGAGAACAAACAGAT |
| Sequence-based reagent | hCYCLIND1_R | This paper | SYBR green PCR primers | TGAGGCGGTAGTAGGACAGG |
| Sequence-based reagent | hTAGLN_F | This paper | SYBR green PCR primers | CGGTTAGGCCAAGGCTCTAC |
| Sequence-based reagent | hTAGLN_R | This paper | SYBR green PCR primers | CCAGCTCCTCGTCATACTTC |
| Sequence-based reagent | haSMA_F | This paper | SYBR green PCR primers | AAGCACAGAGCAAAAGAGGAAT |
| Sequence-based reagent | haSMA_R | This paper | SYBR green PCR primers | ATGTCGTCCCAGTTGGTGAT |
| Sequence-based reagent | hCD44_F | This paper | SYBR green PCR primers | TGGCACCCGCTATGTCGAG |
| Sequence-based reagent | hCD44_R | This paper | SYBR green PCR primers | GTAGCAGGGATTCTGTCTG |
| Sequence-based reagent | hVIM_F | This paper | SYBR green PCR primers | CGAGGAGAGCAGGATTTCTC |
| Sequence-based reagent | hVIM_R | This paper | SYBR green PCR primers | GGTATCAACCAGAGGGAGTGA |
| Sequence-based reagent | hNOTCH3_F | This paper | SYBR green PCR primers | ACCGATGTCAACGAGTGTCT |

*Continued on next page*

*Continued*

| Reagent type (species) or resource | Designation | Source or reference | Identifiers | Additional information |
|---|---|---|---|---|
| Sequence-based reagent | hNOTCH3_R | This paper | SYBR green PCR primers | GTTGACACAG GGGCTACTCT |
| Sequence-based reagent | hZEB2_F | This paper | SYBR green PCR primers | GAGGCGCAAA CAAGCCAATC |
| Sequence-based reagent | hZEB2_R | This paper | SYBR green PCR primers | TCAGAACCTG TGTCCACTAC |
| Sequence-based reagent | hSLUG_F | This paper | SYBR green PCR primers | ACTCCGAAGC CAAATGACAA |
| Sequence-based reagent | hSLUG_R | This paper | SYBR green PCR primers | CTCTCTCTGT GGGTGTGTGT |
| Sequence-based reagent | hFN1_F | This paper | SYBR green PCR primers | CCATAGCTGA GAAGTGTTTTG |
| Sequence-based reagent | hFN1_R | This paper | SYBR green PCR primers | CAAGTACAATCT ACCATCATCC |
| Sequence-based reagent | hVCAM1_F | This paper | SYBR green PCR primers | CGCAAACACTT TATGTCAATGTTG |
| Sequence-based reagent | hVCAM1_R | This paper | SYBR green PCR primers | GATTTTCGGA GCAGGAAAGC |
| Sequence-based reagent | hICAM1_F | This paper | SYBR green PCR primers | TGCCCTGATGG GCAGTCAAC |
| Sequence-based reagent | hICAM1_R | This paper | SYBR green PCR primers | CCCGTTTCAG CTCCTTCTCC |
| Sequence-based reagent | hHPRT1_F | This paper | SYBR green PCR primers | TGAGGATTTG GAAAGGGTGT |
| Sequence-based reagent | hHPRT1_R | This paper | SYBR green PCR primers | TCCCCTGTTG ACTGGTCATT |
| Sequence-based reagent | hCDKN1A_F | This paper | SYBR green PCR primers | CAGCATGACA GATTTCTACC |
| Sequence-based reagent | hCDKN1A_R | This paper | SYBR green PCR primers | CAGGGTATG TACATGAGGAG |
| Sequence-based reagent | hCDKN2A_F | This paper | SYBR green PCR primers | AGCATGGAGCCTTCG |
| Sequence-based reagent | hCDKN2A_R | This paper | SYBR green PCR primers | ATCATGACC TGGATCGG |
| Sequence-based reagent | hIL6_F | This paper | SYBR green PCR primers | GCAGAAAAAG GCAAAGAATC |
| Sequence-based reagent | hIL6_R | This paper | SYBR green PCR primers | CTACATTTG CCGAAGAGC |
| Sequence-based reagent | hIL7_F | This paper | SYBR green PCR primers | TCGATCATT ATTGGACAGC |
| Sequence-based reagent | hIL7_R | This paper | SYBR green PCR primers | AGGAAACACA AGTCATTCAG |
| Sequence-based reagent | hMMP10_F | This paper | SYBR green PCR primers | ACCAATTTATTCCTCGTTGC |
| Sequence-based reagent | hMMP10_R | This paper | SYBR green PCR primers | GTCCGTAGAGAGACTGAATG |
| Sequence-based reagent | hCXCL5_F | This paper | SYBR green PCR primers | ATTTGTCTTGATCCAGAAGC |
| Sequence-based reagent | hCXCL5_R | This paper | SYBR green PCR primers | TCAGTTTTCCTTGTTTCCAC |
| Sequence-based reagent | hANKRD1_F | This paper | SYBR green PCR primers | TGAGTATAAACGGACAGCTC |

*Continued on next page*

*Continued*

| Reagent type (species) or resource | Designation | Source or reference | Identifiers | Additional information |
|---|---|---|---|---|
| Sequence-based reagent | hANKRD1_R | This paper | SYBR green PCR primers | TATCACGGAATTCGATCTGG |
| Sequence-based reagent | hPLAT_F | This paper | SYBR green PCR primers | GGAATTCCATGATCCTGATAG |
| Sequence-based reagent | hPLAT_R | This paper | SYBR green PCR primers | TCCGGCAGTAATTATGTTTG |
| Sequence-based reagent | hPAI-1_F | This paper | SYBR green PCR primers | CGCAACGTGGTTTTCTC |
| Sequence-based reagent | hPAI-1_R | This paper | SYBR green PCR primers | CATGCCCTTGTCATCAATC |
| Sequence-based reagent | hNRARP | This paper | Taqman PCR probes | Hs01104102_S1 |
| Sequence-based reagent | mCdh5 | This paper | Taqman PCR probes | Mm00486938_m1 |
| Sequence-based reagent | mTie1 | This paper | Taqman PCR probes | Mm00441786_m1 |
| Sequence-based reagent | mSerpinH1_F | This paper | SYBR green PCR primers | ATGTTCTTTAAGCCACACTG |
| Sequence-based reagent | mSerpinH1_R | This paper | SYBR green PCR primers | TCGTCATAGTAGTTGTACAGG |
| Sequence-based reagent | mVwa1_F | This paper | SYBR green PCR primers | GATGATCTTC CTATCATTGCC |
| Sequence-based reagent | mVwa1_R | This paper | SYBR green PCR primers | CAATTCCAGCA CGTAGTAAC |
| Sequence-based reagent | mVim_F | This paper | SYBR green PCR primers | CTTGAACGGAAA GTGGAATCCT |
| Sequence-based reagent | mVim_R | This paper | SYBR green PCR primers | GTCAGGCTTGGAAACGTCC |
| Sequence-based reagent | mTgfbr2_F | This paper | SYBR green PCR primers | TCTTTTCGGAAGAATACACC |
| Sequence-based reagent | mTgfbr2_R | This paper | SYBR green PCR primers | GTAGCAGTAGAA GATGATGATG |
| Sequence-based reagent | mVash1_F | This paper | SYBR green PCR primers | CAAGGAAAT GACCAAAGAGG |
| Sequence-based reagent | mVash1_R | This paper | SYBR green PCR primers | ACTGTTGGT GAGGTAAATTC |
| Sequence-based reagent | mSparc_F | This paper | SYBR green PCR primers | GAACCCACATG GCAAGTCTTA |
| Sequence-based reagent | mSparc_R | This paper | SYBR green PCR primers | AAAGCCCAAT TGCAGTTGAGT |
| Sequence-based reagent | mTgfb1_F | This paper | SYBR green PCR primers | CTCCCGTGGCTTCTAGTGC |
| Sequence-based reagent | mTgfb1_R | This paper | SYBR green PCR primers | GCCTTAGTTT GGACAGGATCTG |
| Sequence-based reagent | mApln_F | This paper | SYBR green PCR primers | CAGGCCTATTC CCAGGCTCA |
| Sequence-based reagent | mApln_R | This paper | SYBR green PCR primers | CAAGATCAAG GGCGCAGTCA |
| Peptide, recombinant protein | Recominant human TGF- β | R&D Technologies | 240-B | 50 ng/ml |

*Continued on next page*

*Continued*

| Reagent type (species) or resource | Designation | Source or reference | Identifiers | Additional information |
|---|---|---|---|---|
| Commercial assay or kit | High-Capacity cDNA Reverse Transcription Kit | Applied biosystems | #4368814 | |
| Commercial assay or kit | FastStart Universal SYBR green master mix | Sigma-Aldrich | #04913914001 | |
| Commercial assay or kit | TaqMan gene expression master mix | Applied Biosystems | #4369016 | |
| Commercial assay or kit | SMARTer Stranded Total RNA-Seq Kit V2 – Pico Input Mammalian | Takara Bio, USA | | |
| Commercial assay or kit | SA-β-gal staining kit | Cell signaling technology | #9860 | |
| Commercial assay or kit | Click-iT EdU Alexa Fluor 594 staining kit | Thermo scientific | C10339 | |
| Chemical compound, drug | Hydrogen peroxide | Acros organics | AC202465000 | 200 μM |
| Software, algorithm | Chipster analysis platform (v3.12.2) | CSC, Finland | https://chipster.csc.fi | |
| Software, algorithm | Trimmomatic tool | Chipster, CSC, Finland | https://chipster.csc.fi/manual/trimmomatic.html | |
| Software, algorithm | HISAT2 package | Chipster, CSC, Finland | https://chipster.csc.fi/manual/hisat2.html | |
| Software, algorithm | HTSeq count tool | Chipster, CSC, Finland | https://chipster.csc.fi/manual/htseq-count.html | |
| Software, algorithm | DESeq2 Bioconductor package | Chipster, CSC, Finland | https://chipster.csc.fi/manual/deseq2-pca-heatmap.html | |
| Software, algorithm | PANTHER classification system (V.14.1) | | http://www.pantherdb.org | |
| Software, algorithm | VENNY 2.1 Venn-diagram analysis | BioinfoGP | https://bioinfogp.cnb.csic.es/tools/venny/ | |
| Software, algorithm | MetazSecKB knowledgebase | | http://proteomics.ysu.edu/secretomes/animal/index.php | |
| Software, algorithm | TargetP2.0 server | | http://www.cbs.dtu.dk/services/TargetP/index.php | |
| Software, algorithm | SecretomeP1.0 server | | http://www.cbs.dtu.dk/services/SecretomeP-1.0/ | |
| Software, algorithm | Image J software | NIH, Bethesda | https://imagej.nih.gov/ij/download.html | |
| Software, algorithm | SASP atlas | | http://www.saspatlas.com | |

*Continued on next page*

*Continued*

| Reagent type (species) or resource | Designation | Source or reference | Identifiers | Additional information |
|---|---|---|---|---|
| Software, algorithm | SeneQuest | | https://senequest.net | |
| Software, algorithm | Tabula Muris | | https://tabula-muris.ds.czbiohub.org | |
| Software, algorithm | EndoDB | | https://endotheliomics.shinyapps.io/endodb/ | |
| Software, algorithm | Endothelial cell atlas | | https://endotheliomics.shinyapps.io/ec_atlas/ | |

## Mouse models

All animal experiments were approved by the committee appointed by the District of Southern Finland. Male C57BL/6J wild-type mice were purchased from Janvier Labs and used in the following experimental set-ups: physical activity (progressive exercise training vs sedentary), obesity (high-fat fed for 14 weeks vs chow), aging (18 months vs 2 months), and pressure overload/heart failure (transaortic constriction for 2 and 7 weeks vs sham). Female C57BL/6J wild-type mice of 19–24 months old were used for a separate exercise training experiment. The mice were housed in individually ventilated cages and acclimatized at least for 1 week in the animal facility before any experiments. The cohort size (n) for each experiment is indicated in the figures or figure legends.

## Exercise training

Ten-week-old C57BL/6J male mice (used for RNAseq) or 19–24 months old female mice (used for qPCR analyses) were trained on a treadmill (LE 8710, Bioseb). The mice were familiarized to the treadmill for three consecutive days with low speed (8–10 cm/s). Progressive training program consisted of 1–1.5 hr training bouts 5 days a week for a total of 6 weeks with increasing speed, inclination, and/or duration each week. The following parameters in the treadmill controller were opted, tread inclination: 0°−10°; minimum and maximum tread speed: 10–30 cm/s; shock grid intensity: 0.2 mA. The aged female mice were exercise-trained for 4 weeks and the same procedures were followed during the training program.

## High fat feeding

Ten-week-old C57BL/6J male mice were fed with standard chow diet or HFD containing 60% kcal derived from fat (Research Diets, D12492) for 4 or 14 weeks and used for immunohistochemistry or RNA-seq analysis, respectively.

## TAC surgery

Ten-week-old C57BL/6J male mice were anesthetized with ketamine and xylazine. The mice were placed in supine position and intubated. The skin along the supra-sternal notch to mid sternum was incised to perform sternotomy to expose the aortic arch, right innominate, and left common carotid arteries together with the trachea. Ligation of the transverse aorta between the right innominate left common carotid arteries against blunted 27-gauge needle with a 7–0 suture was performed and the needle was gently removed. The sternum and skin were ligated with monofilament polypropylene suture. Mice were placed in a warm chamber to recover, treated with analgesics (0.05 mg/kg of Temgesic i.m.) at the time of the surgery and twice a day for following 2 days. For the control group (sham), all the steps in the surgical procedure were followed, except constricting the aorta. One group was killed 2 weeks and another group 7 weeks after the surgery. Echocardiography was performed once a week during the experiment.

## Echocardiography

To analyze cardiac function and ventricle dimensions, two-dimensional echocardiography images were acquired (Vevo 2100 Ultrasound, FUJIFILM Visual Sonics). The LV internal diameter, LV posterior wall thickness, and interventricular septum thickness at end-systole and end-diastole were

measured in M-mode along the parasternal short axis view and analyzed by Simpson's modified method (*Kivelä et al., 2019*).

## Body fat measurement

The mice were anesthetized with ketamine and xylazine and the percentage of total body fat was measured using dual energy X-ray absorptiometry (Lunar PIXImus, GE Medical systems).

## Oral glucose tolerance test

Mice were fasted for 4–5 hr before the experiment. Glucose (1 g/kg) was administered by oral gavage to mice. Blood from the tail tip was used to measure glucose levels at the following time points (15, 30, 60, and 90 min) using blood glucose meter (Contour, Bayer).

## Immunofluorescent staining

Frozen mouse heart sections (10 µm) were cut with cryomicrotome and stained as described previously (*Kivelä et al., 2019*). The primary antibodies are listed in the Key Resource Table. Primary antibodies were detected with Alexa 488, 594 or 647-conjugated secondary antibodies (Molecular Probes, Invitrogen). The sections were mounted with Vectashield Hard Set mounting media with DAPI (Vector Laboratories). The images were acquired with 20×, 40× air or 40× oil immersion objectives using AxioImager epifluorescent microscope (Carl Zeiss). The stained micrographs were initially adjusted for threshold, and an area fraction tool was used to quantify the area percentage of the vessels and collagen (Image J software, NIH).

## Human heart samples

Human heart samples were obtained from four organ donor hearts, which could not be used for transplantation for example due to size or tissue-type mismatch. The collection was approved by institutional ethics committee and The National Authority for Medicolegal Affairs.

## Immunohistochemistry

The human paraffin heart sections (4 µm) were cut, deparaffinized, and rehydrated with xylene, descending concentration series of ethanol (99%, 95%, 70%, and 50%) and $H_2O$, and incubated in high pH antigen retrieval buffer containing 10 mM Tris, 1 mM EDTA, 0.05% Tween 20 (pH 9.0). For HSP47 immunohistochemical analysis, VECTASTAIN Elite ABC kit (PK-6100) and DAB substrate were used to label and amplify the antibody signal. The 20× or 63× images were acquired with light microscope (Leica). For immunofluorescent staining, after the antigen retrieval step the sections were blocked with donkey immunomix (5% normal donkey serum, 0.2% BSA, 0.3% Triton X-100 in PBS), incubated overnight at 4°C with the primary antibodies for HSP47 and VE-Cadherin (CDH5) and detected with Alexa 488 and 594 conjugated secondary antibodies (Molecular probes, Invitrogen). The sections were mounted with Vectashield hardset with DAPI (Vector labs) and 40× images were acquired using AxioImager epifluorescent microscope (Carl Zeiss).

## Isolation of cardiac ECs

The harvested hearts were briefly rinsed in ice-cold Dulbecco's phosphate-buffered saline (DPBS, #14190–094, Gibco) supplemented with 0.3 mM calcium chloride ($CaCl_2$), cut opened longitudinally into two halves to expose the cardiac chambers and minced longitudinally and transversely into small pieces. To enzymatically dissociate the heart, 4 ml of pre-warmed digestion media (1 mg/ml) of each collagenase types (type I [#17100–017], type II [#17101–015], and type IV [#17104–019]) from Gibco were dissolved in DPBS containing 0.3 mM $CaCl_2$ and added to the minced hearts, and incubated in water bath at 37°C for 25 min. During the digestion process, the samples were very gently mixed by vortexing for every 5 min. After incubation, the cell suspension was gently passed through T10 serological pipette 20 times. To neutralize the digestion, 10 ml of rinsing media (Dulbecco's modified eagle medium [#31053–028] supplemented with 10% heat inactivated FCS) was added to the cell suspension and filtered through the 70 µm nylon cell strainer (Corning, #352350). Throughout the isolation process the cell suspensions were centrifuged for 5 min, 300 g, and 4°C between each rinsing step. The cell pellet was resuspended in 5 ml of ice-cold staining buffer (DPBS containing 2% heat inactivated FCS and 1 mM EDTA). Before antibody staining, the cells were incubated with Fc

receptor blocking antibody (CD16/32) for 5 min. The cells were incubated with the CD31, PDGFRa/CD140a, CD45, and Ter119 antibodies for 30 min (Key Resource Table for the antibody details). Prior to FACS, the cells were rinsed twice with the staining buffer and filtered through 5 ml cell strainer tubes (Corning, #352235).

## Fluorescent-activated cell sorting

The cells were passed through a 100 μm nozzle. Multiple light scattering parameters for forward- and side-scatter properties of the cells were employed to gate, analyze, and sort live cardiac ECs. Initially, total cells were gated based on the forward and side-scatter area of the cells (FSC-A and SSC-A). The single cells were selected depending on forward scatter parameters area, height, and width of the cells (FSC-A, FSC-H, or FSC-W). DAPI was used to determine live and dead cells. To enrich and FACS sort pure and viable cardiac ECs, ECs were stained with CD31, mesenchymal cells with PDGFRa/CD140a, leucocytes with CD45, and red blood cells with Ter119. The live cardiac ECs were defined as CD31$^+$ CD45$^-$ Ter119$^-$ CD140a$^-$ DAPI$^-$. Cells were sorted using FACS Aria II (BD Biosciences), and the data was acquired with BD FACSDIVA v8.0.1 and further analyzed with FlowJo v10.1 (FlowJo, LLC) software. We verified the enrichment and purity of the FACS sorted Cardiac EC population (CD31+ PDGFRa (CD140a)- CD45- Ter119- DAPI-) by QPCR analysis for classical cardiac EC markers. Recently, we have used the same isolation method for single-cell RNAseq experiments, and these results show that there is about 3% contamination from other cells types, mainly pericytes and hemangioblasts.

## RNA isolation

The sorted cardiac ECs were immediately suspended in lysis buffer (350 μl of RLT buffer plus 10 μl of β-mercaptoethanol), the cells were homogenized in QIAshredder (#79654, Qiagen), and the RNA was purified using RNeasy Plus Micro Kit (#74034, Qiagen) according to the manufacturer's instruction. The RNA integrity was analyzed with bioanalyzer (Agilent Tape Station 4200) and the concentration was determined by Qubit fluorescence assay (ThermoFisher). The cells from the post sort fractions were stained with propidium iodide (PI) and the viability of the cells were determined by Luna automated cell counter. The purity of the post sort fraction was determined by QPCR analysis for EC markers.

## RNA sequencing of cardiac EC

Indexed cDNA library was synthesized using SMARTer Stranded Total RNA-Seq Kit V2 – Pico Input Mammalian (Takara Bio, USA) kit according to the manufacturer's instructions. The library quality was determined using bioanalyzer and sequenced using illumina NextSeq 550 System with the following specifications: 1 × 75 bp, 50M single end reads were sequenced using NextSeq 500/550 High-Output v2.5 kit.

## Differential gene expression

The sequenced reads were analyzed with the following software packages embedded in the Chipster analysis platform (*Kallio et al., 2011*) (v3.12.2; https://chipster.csc.fi). Trimmomatic tool (*Bolger et al., 2014*) (https://chipster.csc.fi/manual/trimmomatic.html) was used to preprocess Illumina single end reads. The HISAT2 package (*Kim et al., 2015*) (https://chipster.csc.fi/manual/hisat2.html) was employed to align the reads to mouse genome GRCm38.90 and the HTSeq count tool (*Anders et al., 2015*) (https://chipster.csc.fi/manual/htseq-count.html) to quantify the aligned reads per gene. The raw read count table for genes generated utilizing the HTSeq count were used as an input to perform two-dimensional principal component analysis (PCA) and unsupervised hierarchical clustering analysis using DESeq2 Bioconductor package (*Love et al., 2014*) (https://chipster.csc.fi/manual/deseq2-pca-heatmap.html). Next, to perform the differential gene expression (DGE) analysis, the DESeq2 Bioconductor package (*Love et al., 2014*) was used. The advantage of DEseq2 tool is sensitive and precise for analyzing the DEG in studies with few biological replicates. To reliably estimate the within group variance, Empirical Bayes shrinkage for dispersion estimation was used and a dispersion value for each gene was estimated through a model fit procedure (refer to the *Figure 2—figure supplement 2A*, which illustrates the shrinkage estimation for the experimental conditions). The gene features obtained after the dispersion estimation were used to perform statistical

testing. Next, negative binomial generalized linear model was fitted for each gene and Wald test (raw p-value) was calculated to test the significance. Finally, DEseq2 applies Benjamini–Hochberg correction test to control the FDR (refer to the *Figure 2—figure supplement 2B* indicating the distribution of raw and FDR adjusted p-value for the experimental conditions). In our DEG analysis, we have set the FDR (p adj.) cutoff as less than or equal to 0.05 (FDR/p-adj $\leq$ 0.05) for pathway analysis and gene overlap analysis. The RNA sequencing data is deposited in the GEO database, under the series accession number GSE145263.

## Gene function and pathway analysis

The gene function and pathway analysis of the DGE were determined by performing statistical over-representation test using the PANTHER classification system (*Mi et al., 2019*) (V.14.1; http://www.pantherdb.org). The p<0.05 was considered for the further analysis and the data is presented as $-\log_2$(p-value).

## Gene overlap and in silico gene characterization

The differentially expressed up- and downregulated genes (adjusted p-value 0.05) from the different experimental conditions were imported to VENNY 2.1 Venn-diagram analysis software (BioinfoGP; https://bioinfogp.cnb.csic.es/tools/venny/) to identify genes which were significantly affected by several experimental conditions. The MetazSecKB knowledgebase (*Meinken et al., 2015*) (http://proteomics.ysu.edu/secretomes/animal/index.php), TargetP2.0 server (*Almagro Armenteros et al., 2019*) (http://www.cbs.dtu.dk/services/TargetP/index.php), and SecretomeP1.0 server (*Bendtsen et al., 2004*) (http://www.cbs.dtu.dk/services/SecretomeP-1.0/) were used to characterize molecular functions, subcellular localizations, and possible secretion properties of the identified common genes.

## Cell culture and lentiviral production

HUVEC and HCAEC were purchased from PromoCell (cell lines were authenticated and tested for mycoplasma status by the vendor). Cell cultures in the lab are regularly checked for their mycoplasma status using Mycoalert mycoplasma detection kit (LT07-218, Lonza). Both HUVEC and HCAEC were cultured and maintained in EC growth Basal Medium MV (C-22220, PromoCell) supplemented with Supplement Pack GM MV (C-39220, PromoCell) and gentamycin. For both gene overexpression and silencing studies, 80% confluent monolayer culture of HUVECs and HCAECs was used.

To overexpress SERPINH1 in EC, we cloned a lentiviral vector FUW-hSERPINH1-Myc (map and plasmid available by request). A scrambled sequence in the same vector was used as a control. 293FT cells (ATCC) were cultured and maintained in DMEM supplemented with 10% FCS and L-glutamine, and co-transfected with the lentiviral packaging plasmid vectors CMVg, CMV$_\Delta$8.9, and the target plasmid. The supernatants were collected at 48 and 72 hr, and concentrated by ultracentrifugation as described previously (*Lois et al., 2002*). For overexpression, HUVEC and HCAEC were transfected with lentivectors for 48 hr. For gene silencing studies, HCAEC was treated with lentivectors encoding for four independent clones of human shSERPINH1 for 24 hr. Subsequently, the cells were treated with puromycin (2 µg/ml) for 48 hr to select the transduced cells. After selection, the cells were used for further analysis. The clone id and target sequence for human shSERPINH1 constructs are shown in the Key Resource Table.

## Scratch wound assay

The SERPINH1 overexpressed or silenced HCAECs were seeded in the IncuCyte ImageLock 96-well microplate precoated with 0.1% gelatin and cultured in complete EC growth medium. To the confluent cell monolayers, 700–800 micron scratch wounds were introduced with IncuCyte WoundMaker, and the wells were briefly rinsed with and maintained in complete EC growth medium. The kinetics of the cell migration were recorded and 10× phase contrast time-lapse images were acquired using IncuCyte Live-Cell Analysis System. The wound closure region was measured by Edge-detection and thresholding method in Image J software (NIH). The data is presented as wound closure (%) relative to time.

## EndMT assay

The coverslips or 6-well plates were precoated with 0.1% gelatin for 20 min at 37°C, scrambled or SERPINH1 silenced HCAEC were seeded and cultured in complete EC growth medium. The cells were treated with or without 50 ng/ml of recombinant human TGF-β (R and D Technologies) and/or 200 μM hydrogen peroxide (Acros organics) for 5 days as described previously (*Evrard et al., 2016*; *Magenta et al., 2011*).

## Cell staining

The cells grown on the coverslips were fixed with 4% PFA in PBS for 15 min. Blocking was done using donkey immunomix and the cells were stained with primary antibodies and secondary antibodies as indicated in the Key resources table. DAPI was used to stain the nucleus, and the cells were mounted using Vectashield (Vector labs). The amount of COL1 was quantified by adjusting 10× images for threshold and area fraction tool was used to quantify the area percentage of the collagen deposition (Image J software, NIH).

## SA-β-gal staining

The SERPINH1 overexpressed HCAECs at passage 6 (P6) were seeded on the coverslips coated with 0.1% gelatin. The cells were allowed to reach 80% confluence, rinsed twice with ice cold PBS, incubated in the fixative solution (#11674, Cell signaling technology) at room temperature for 10 min, rinsed twice with ice cold PBS, and stored at 4°C. The senescence-associated beta-galactosidase (SA-β-gal) activity at pH 6.0 was detected with the SA-β-gal staining kit (#9860, Cell Signaling Technology) according to the manufacturer's instructions. The SA-β-gal+ cells were quantified using point tool (Image J software, NIH) and normalized to the total number of cells per field. The data presented as percentage of SA-β-gal+ cells of all cells.

To detect SA-β-gal activity in the hearts of HFD- and chow-fed mice, 4 μm thick cryo sections were fixed with 1% PFA in PBS for 1 min at room temperature, and the sections were rinsed twice with PBS and incubated in the β-galactosidase staining solution, pH 6.0 (#9860, Cell signaling technology) for 24 hr at 37°C. The slides were rinsed twice with PBS, post fixed with 1%PFA in PBS for 1 min, rinsed twice with PBS, counter stained with 0.1% eosin, rinsed twice with distilled water, and the sections were mounted with Immuno-mount (Thermo scientific) (*Cazin et al., 2017*). The images were acquired using light microscope (Leica) and the SA-B-gal+ cells were scored using point tool (Image J software, NIH) per field. The data is presented as SA-β-gal+ cells/field.

## Edu incorporation assay

The SERPINH1 overexpressed and silenced HCAECs were seeded on the coverslips coated with 0.1% gelatin and cultured in complete EC growth medium overnight. The cells were allowed to reach 70% confluence, incubated with 10 μM of EdU labeling solution in EC complete growth media for 7 hr under normal culture conditions. The cells were rinsed twice with ice cold PBS and fixed with 4% PFA in PBS for 10 min. The manufacturer's instructions in the Click-iT EdU Alexa Fluor 594 staining kit (Thermo scientific) were followed to detect the Edu+ proliferating cells and the nuclei were counterstained with Hoechst. The percentage of Edu+ cells were normalized to Hoechst+ nuclei using area fraction tool (Image J software, NIH). The data is presented as percentage of EdU/Hoechst (%).

## Analysis of SASP gene expression

To analyze the expression of SASP genes in our cardiac EC RNA-seq data sets (HFD and TAC models), we have reviewed and compared the DEGs in our data sets with FDR threshold of 0.05 using the following databases: SASP atlas (http://www.saspatlas.com) (*Basisty et al., 2020*) and Sene-Quest (https://senequest.net). Further, the endothelial expression of the identified genes was checked using Tabula Muris database (https://tabula-muris.ds.czbiohub.org).

## Real-time quantitative PCR

RNA from the cultured cells was purified and isolated using NucleoSpin RNA II Kit according to the manufacturer's protocol (Macherey-Nagel). cDNA was synthesized with High-Capacity cDNA Reverse Transcription Kit (Applied biosystems, #4368814). SYBR green or TaqMan gene expression

assays were performed using FastStart Universal SYBR green master mix (Sigma-Aldrich, #04913914001) and TaqMan gene expression master mix (Applied Biosystems, #4369016), respectively. mRNA expression was analyzed using Bio-Rad C1000 thermal cycler according to standardized protocol of the qPCR master mix supplier. The average of the technical triplicates for each sample was normalized to the housekeeping gene HPRT1. The mRNA expression levels were calculated and presented as fold change (Ctrl = 1). The primer sequences are listed in the Key Resource Table.

### Western blotting

The cells were harvested and homogenized in lysis buffer containing 0.5% NP-40 (v/v) and 0.5%Triton X-100 (v/v) in PBS, supplemented with protease and phosphatase inhibitors (A32959, Pierce, Thermo Scientific). Protein concentration was determined using a BCA protein assay kit (Pierce, Thermo Scientific). Equal amounts of total protein were resolved in Mini-PROTEAN TGX Precast gels (Bio-Rad) and transferred to PVDF membrane (immobilon-P, Millipore). 5% BSA (wt/vol) and 0.1% Tween 20 (v/v) in TBS were used to block the membranes followed by incubation with primary antibodies listed in the (Key Resource Table) overnight at 4°C. HRP-conjugated secondary antibodies (DAKO) were used, and HRP signals were developed with Super-Signal West Pico Chemiluminescent substrate or Femto Maximum sensitivity substrate (Thermo Scientific). The blots were imaged with Odyssey imager (Li-COR Biosciences) or Chemi Doc imaging system (Bio-Rad) and quantified with Image Studio Lite Software (Li-COR Biosciences).

### Statistics

The data from the individual experiments were analyzed by Student's t-test. $p < 0.05$ value was considered statistically significant and p-values in the graphs are shown as $*p < 0.05$, $**p < 0.01$, and $***p < 0.001$. The data is shown as mean ± SEM. The GraphPad Prism 7 software was used for statistical analysis.

## Acknowledgements

We would like to thank Dr. Ralf Adams and Dr. Guillermo Luxán for their help in setting up the EC isolation; Dr. Seppo Kaijalainen for cloning the SERPINH1/HSP47 overexpression vector; and Kirsi Mattinen, Päivi Leinikka, Maria Arrano de Kivikko, Ilse Paetau, Tanja Laakkonen, and Tapio Tainola for their excellent technical help. We thank the Laboratory Animal Center, the Biomedicum Imaging Unit, the HiLife Flow Cytometry Unit, the Biomedicum Functional Genomics Unit, and the AAV Gene Transfer and Cell Therapy Core Facility for the help and facilities.

## Additional information

### Funding

| Funder | Grant reference number | Author |
|---|---|---|
| Jenny ja Antti Wihurin Rahasto | | Karthik Amudhala Hemanthakumar Riikka Kivelä |
| Academy of Finland | 297245 | Riikka Kivelä |
| Sydäntutkimussäätiö | | Karthik Amudhala Hemanthakumar Riikka Kivelä |
| Sigrid Juséliuksen Säätiö | | Riikka Kivelä |
| Suomen Kulttuurirahasto | | Riikka Kivelä |
| Suomen Lääketieteen Säätiö | | Mikko I Mäyränpää |
| Biomedicum Helsinki-säätiö | | Karthik Amudhala Hemanthakumar |
| Aarne Koskelon Säätiö | | Karthik Amudhala Hemanthakumar |

The funders had no role in study design, data collection and interpretation, or the decision to submit the work for publication.

## Author contributions

Karthik Amudhala Hemanthakumar, Conceptualization, Data curation, Formal analysis, Investigation, Visualization, Methodology, Writing - original draft, Writing - review and editing; Shentong Fang, Andrey Anisimov, Formal analysis, Methodology; Mikko I Mäyränpää, Resources; Eero Mervaala, Methodology; Riikka Kivelä, Conceptualization, Resources, Supervision, Funding acquisition, Methodology, Writing - original draft, Project administration, Writing - review and editing

## Author ORCIDs

Karthik Amudhala Hemanthakumar [ID] https://orcid.org/0000-0001-6151-1005
Shentong Fang [ID] http://orcid.org/0000-0002-3520-7007
Andrey Anisimov [ID] http://orcid.org/0000-0003-0259-1273
Riikka Kivelä [ID] https://orcid.org/0000-0002-2686-8890

## Ethics

Human subjects: Human heart samples were obtained from 4 organ donor hearts, which could not be used for transplantation e.g. due to size or tissue-type mismatch. The collection was approved by institutional ethics committee and The National Authority for Medicolegal Affairs.
Animal experimentation: All animal experiments were approved by the committee appointed by the District of Southern Finland (permit number ESAVI/22658/2018). The study was performed in accordance with the recommendations of FELASA. All of the animals were handled according to approved institutional animal care and use committee of the University of Helsinki. All surgery was performed under anesthesia advised by the University's veterinarians, and every effort was made to minimize suffering.

## Decision letter and Author response

Decision letter https://doi.org/10.7554/eLife.62678.sa1
Author response https://doi.org/10.7554/eLife.62678.sa2

# Additional files

## Supplementary files

• Transparent reporting form

## Data availability

All RNA sequencing data have been deposited in GEO under accession code GSE145263.

The following dataset was generated:

| Author(s) | Year | Dataset title | Dataset URL | Database and Identifier |
|---|---|---|---|---|
| Kivelä R, Hemanthakumar KA | 2020 | RNA sequencing of cardiac endothelial cells from the cardiovascular disease risk factor mouse models | https://www.ncbi.nlm.nih.gov/geo/query/acc.cgi?acc=GSE145263 | NCBI Gene Expression Omnibus, GSE145263 |

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
