## [Decision Letter]

**Acceptance summary:**

The manuscript utilizes a transcriptomic approach to identify a common mechanism underlying endothelial dysfunction induced by several cardiovascular disease risk factors. Based on leads from the transcriptomics data, the authors performed histology, in vitro gene overexpression, and silencing studies to show that TGFβ signaling mediated the upregulation of a serpin that in turn mediates endothelial to mesenchymal transition in cardiac endothelial cells. It is timely work that integrates the effects of risk factors horizontally and also examines function for some of the noted changes, and thus it provides a more thorough understanding of age-related cardiovascular disease.

**Decision letter after peer review:**

Thank you for submitting your article "Cardiovascular disease risk factors induce mesenchymal features and senescence in cardiac endothelial cells" for consideration by *eLife*. Your article has been reviewed by three peer reviewers, and the evaluation has been overseen by a Reviewing Editor and Jessica Tyler as the Senior Editor. The following individuals involved in review of your submission have agreed to reveal their identity: Peetra Magnusson (Reviewer #1); Priya Balasubramanian (Reviewer #2).

The reviewers have discussed the reviews with one another and the Reviewing Editor has drafted this decision to help you prepare a revised submission.

Summary:

The manuscript by Hemanthakumar et al. titled “Cardiovascular disease risk factors induce mesenchymal features and senescence in cardiac endothelial cells” utilized a transcriptomic approach to identify a common mechanism underlying endothelial dysfunction induced by several CVD risk factors. Based on leads from the transcriptomics data, the authors performed histology and in vitro gene overexpression and silencing studies to show that TGFβ signaling mediated upregulation of SERPINH1/Hsp47 mediates endothelial to mesenchymal transition in cardiac endothelial cells. Overall, the paper is well-written with extensive data that support the conclusions. It is timely work that integrates effects of risk factors horizontally and also examines function for some of the noted changes, and thus it provides a more thorough understanding of age-related CVD.

Reviewers were in agreement that a modest amount of validation of senescence phenotype would strengthen the conclusions. It was suggested to consider RT-PCR, senescence-associate b-gal in vitro, and/or perhaps investigating expression of SASP (senescence-associated-secretory phenotype) genes. This was thought to be important for obesity and pressure overload groups, that have not been linked to EC senescence in the literature. For example, subsection “SERPINH1/HSP47 is needed for collagen 1 deposition by ECs”, larger cells in HSP47 overexpression, is this characterized by senescence and could it be investigated by b-galactosidase staining.

Revisions expected in follow-up work:

If you cannot perform some of the suggested experiments due to pandemic-related issues, we ask that you temper your conclusions regarding the link to senescence for this work, and plan to submit further work as described above.

Numerous points were raised by the reviewers that it was felt could be dealt with via text revisions and/or data reanalysis:

1) Please discuss what might drive repression of TGF-b upon exercise? Removal of stress products, better oxygenation, improved flow and thereby reduced EndMT?

2) How do markers of inflammation change in the datasets? Note that TGFb is also linked to inflammation.

3) What is the potential effect of sex-specific changes? One reviewer noted that some of the groups were not sex-matched, and it was felt that discussion of this variable was warranted. For example, while most experimental approaches used male mice, females were used for the exercise training experiment in old mice. They were also older that the males used in the aging approach. Remarkably, exercise suppressed Vwa1 and Tgfbr2 expression in cardiac EC from young male but not from aged female whereas exercise diminished Vim expression in EC from aged female but not from young males. Is it possible that sex was a confounding factor for these discrepancies? Please clearly state the sex throughout the text (and figures) and also emphasize that sex needs to be take into account on their conclusion.

4) Please discuss HSP47 expression – it was expressed throughout the human cardiac EC. Differential expression in arteries vs veins, what about specifically the coronary vessels?

5) Please speculate a little more on relevance of genes in different groups – for example, Figure 3: Are all genes presented that are up in sedentary then "bad" genes? How come exercise creates increased downregulation of genes while fewer upregulated? Does exercise provide more of a break upon gene regulation than actually stimulating the transcriptome? Is exercise more suppressive than promoting? Or just skewed presentation of data?

6) Figure 6B, lentiviral silencing of Serpinh1, batch #3 and #4 affected cell viability? Authors claim decreased cell density. Or is proliferation affected i.e. downregulated?

7) It is counter intuitive to see that overexpression of SERPINH1 promotes cell proliferation in wound healing assay when it is associated with senescence wherein proliferative arrest is expected. Please address this in the Discussion.

8) In Figure 1B, please confirm the number of animals in TAC(2) group. It appears that there are only 2 animals in the group which makes it not ideal for statistical analysis.

9) Although the expression of SerpinH1 was validated in human hearts, healthy hearts were used for that approach. Is there any evidence that SerpinH1 is increased in patients presenting the major risks factors for CVD? Perhaps considering citing Kato et al., 2017.

---

## [Author Response]

Revisions for this paper:Reviewers were in agreement that a modest amount of validation of senescence phenotype would strengthen the conclusions. It was suggested to consider RT-PCR, senescence-associate b-gal in vitro, and/or perhaps investigating expression of SASP (senescence-associated-secretory phenotype) genes. This was thought to be important for obesity and pressure overload groups, that have not been linked to EC senescence in the literature. For example, subsection “SERPINH1/HSP47 is needed for collagen 1 deposition by ECs”, larger cells in HSP47 overexpression, is this characterized by senescence and could it be investigated by b-galactosidase staining.

We thank the reviewers and the editors for positive and constructive comments on our manuscript. We have now performed all the suggested revision experiments for the paper as follows:

SA-β galactosidase staining and qPCR for SASP genes to determine the effect of SERPINH1 overexpression on EC senescence. We found that SERPIH1 induced significantly increased expression of SASP genes and SA-β-gal positive cells, and this data is added to the Figure 5G-H.Analysis of SASP gene expression in obesity and pressure overload experiments. Both CVD risk factors induced the expression of several SASP genes, and this data is now included in the Figure 3—figure supplement 2A-E.Analysis of inflammatory gene expression in aging, obesity, pressure overload and exercise models. The inflammatory gene signature is induced in all of the CVD risk factor groups, whereas several inflammation-related genes were repressed by exercise. The heat maps are presented in the Figure 3—figure supplement 1A-E.EdU labeling experiment to determine the effect of SERPINH1 overexpression and silencing on human EC proliferation. Overexpression was found to modestly increase proliferation, whereas silencing almost completely inhibited proliferation. The data is added to the Figure 7E-G.

We have also modified the Discussion based on the new results, detailed in the following responses.

Numerous points were raised by the reviewers that it was felt could be dealt with via text revisions and/or data reanalysis:1) Please discuss what might drive repression of TGF-b upon exercise? Removal of stress products, better oxygenation, improved flow and thereby reduced EndMT?

There is no mechanistic data on the effects of exercise on TGF-b, however, it has been shown that NO is a potent negative regulator of TGF-b/Smad activity in ECs (PMID: 16239590). Exercise increases blood flow and shear stress, inducing eNOS expression and NO production, thus possibly providing link to attenuated TGF-b pathway activity in our experiment. Interestingly, there are recent papers linking exercise to TGF-b signaling in rat heart and human skeletal muscle. We have included discussion about this.

2) How do markers of inflammation change in the datasets? Note that TGFb is also linked to inflammation.

We have now re-analyzed the in vivo RNAseq data sets for the expression of inflammatory genes, and the results are presented in Figure 3—figure supplement 1A-E for all data sets. The genes were selected based on a recent paper by Mike Simons’ group published in Nature Metabolism, where they present a gene list for TGF-b mediated inflammatory response in ECs (PMID: 31572976). We also used Gene Ontology term “inflammatory response” (GO:0006954) for validating the genes for the analysis.

3) What is the potential effect of sex-specific changes? One reviewer noted that some of the groups were not sex-matched, and it was felt that discussion of this variable was warranted. For example, while most experimental approaches used male mice, females were used for the exercise training experiment in old mice. They were also older that the males used in the aging approach. Remarkably, exercise suppressed Vwa1 and Tgfbr2 expression in cardiac EC from young male but not from aged female whereas exercise diminished Vim expression in EC from aged female but not from young males. Is it possible that sex was a confounding factor for these discrepancies? Please clearly state the sex throughout the text (and figures) and also emphasize that sex needs to be take into account on their conclusion.

We thank the reviewers pointing this out. We used male mice for all RNAseq experiments. Following these findings, we also wanted to test the effect of exercise in old mice, especially to see if it can attenuate SerpinH1 expression also in old mice. For this experiment, we chose female mice to evaluate, if the effect is seen in females as well. The perfect set up would have included both old male and female mice with and without exercise training, however, in practice this was not feasible. Even though exercise training attenuated SerpinH1 expression also in aged female mice, not all genes responded the same as in young males. Thus, it will be important to evaluate the effects of CVD risk factors also in female mice, both young and old. We have now added information about the sex of the mice to all figure legends and clarified this in the Materials and methods, and added the sex aspect also to the Discussion section.

4) Please discuss HSP47 expression – it was expressed throughout the human cardiac EC. Differential expression in arteries vs veins, what about specifically the coronary vessels?

We used EndoDB (dataset E-GEOD-43475) to compare the HSP47 expression in various human endothelial cells, both cell lines and primary cells. The results show that there is abundant and highly similar expression of SERPINH1/HSP47 in endothelial cells from different tissues and in both arteries and veins. The results are added as a Figure 4—figure supplement 3A and to the Results and Discussion sections.

5) Please speculate a little more on relevance of genes in different groups – for example, Figure 3: Are all genes presented that are up in sedentary then "bad" genes? How come exercise creates increased downregulation of genes while fewer upregulated? Does exercise provide more of a break upon gene regulation than actually stimulating the transcriptome? Is exercise more suppressive than promoting? Or just skewed presentation of data?

Thank you for pointing this out, we have now added more detailed discussion in the text. We believe that the reason why exercise had more downregulated that upregulated genes is due to the chronic effects that were studied here. Acute exercise would likely induce e.g. angiogenesis genes, which are already attenuated after six weeks of training, when the vasculature has been already adapted to the training load. Thus, at this point the upregulated genes reflect quiescence and stabilization of the vasculature and downregulated genes attenuation of (pathological) signaling pathways, e.g. TGF-b.

6) Figure 6B, lentiviral silencing of Serpinh1, batch #3 and #4 affected cell viability? Authors claim decreased cell density. Or is proliferation affected i.e. downregulated?

We have now performed EdU labeling to analyze the proliferation vs. migration effects of SERPINH1 overexpression and silencing. Overexpression significantly increases EC proliferation about 20%, however, this is much smaller effect than seen in the wound healing assay. Based on this, we conclude that SERPINH1 increases especially cell migration, which is in line with increased mesenchymal properties. In contrast, silencing of SERPINH1 resulted in almost complete inhibition of proliferation, which explains the loss of cells after 10 days in culture, and also the impaired wound healing. These new results are added to Figure 7E-G and to the Results section.

7) It is counter intuitive to see that overexpression of SERPINH1 promotes cell proliferation in wound healing assay when it is associated with senescence wherein proliferative arrest is expected. Please address this in the discussion.

This is true, and we have now looked at this in more detail during the revision. Indeed, the analysis of SA-β-gal and SASP genes confirms the pathway analysis results, that SERPINH1 promotes senescence in ECs. Counter intuitively, the EdU analysis demonstrated slightly increased proliferation, but together with the wound healing assay the results suggest that increase in migration is more pronounced. We have now discussed this discrepancy and propose that the increased migration and proliferation reflect the acquired mesenchymal features, which likely overrides the effects of senescence-related changes, at least in early cultures (48h post-transduction), but this might change over time.

8) In Figure 1B, please confirm the number of animals in TAC(2) group. It appears that there are only 2 animals in the group which makes it not ideal for statistical analysis.

For the FACS analysis of this group, we unfortunately lost two samples. However, we have reported the two samples as they are highly similar to the ones observed at 7 weeks after TAC. Thus, even this is not enough for reliable statistical analyses, we believe that these two values represent the phenomenon rather well.

9) Although the expression of SerpinH1 was validated in human hearts, healthy hearts were used for that approach. Is there any evidence that SerpinH1 is increased in patients presenting the major risks factors for CVD? Perhaps considering citing Kato et al., 2017.

Currently, there is not much data about SERPINH1/HSP47 in human disease, especially related to CVD. The study by Kato et al. proposes that EndMT and increased SERPINH1/HSP47 are related to atrial fibrillation, which is often related to fibrosis. We have now discussed this in the text.